# Learning to Iteratively Solve Routing Problems with Dual-Aspect Collaborative Transformer

**Yining Ma[1], Jingwen Li[1], Zhiguang Cao[2,*], Wen Song[3,*], Le Zhang[4],**
**Zhenghua Chen[5], Jing Tang[6]**

[1]National University of Singapore
[2]Singapore Institute of Manufacturing Technology, A*STAR
[3]Institute of Marine Science and Technology, Shandong University
[4]University of Electronic Science and Technology of China
[5]Institute for Infocomm Research, A*STAR
[6]The Hong Kong University of Science and Technology
{yiningma, lijingwen}@u.nus.edu, zhiguangcao@outlook.com,
wensong@email.sdu.edu.cn, zhangleuestc@gmail.com,
chen0832@e.ntu.edu.sg, jingtang@ust.hk

## Abstract

Recently, Transformer has become a prevailing deep architecture for solving vehicle routing problems (VRPs). However, it is less effective in learning *improvement* models for VRP because its positional encoding (PE) method is not suitable in representing VRP solutions. This paper presents a novel *Dual-Aspect Collaborative Transformer* (DACT) to learn embeddings for the node and positional features separately, instead of fusing them together as done in existing ones, so as to avoid potential noises and incompatible correlations. Moreover, the positional features are embedded through a novel *cyclic positional encoding* (CPE) method to allow Transformer to effectively capture the circularity and symmetry of VRP solutions (i.e., cyclic sequences). We train DACT using Proximal Policy Optimization and design a curriculum learning strategy for better sample efficiency. We apply DACT to solve the traveling salesman problem (TSP) and capacitated vehicle routing problem (CVRP). Results show that our DACT outperforms existing Transformer based improvement models, and exhibits much better generalization performance across different problem sizes on synthetic and benchmark instances, respectively.

## 1 Introduction

Vehicle Routing problems (VRPs), such as the Traveling Salesman Problem (TSP) and the Capacitated Vehicle Routing Problem (CVRP) which consider finding the optimal route for a single or fleet of vehicles to serve a set of customers, have ubiquitous real-world applications [1, 2]. Despite being intensively studied in the Operations Research (OR) community, VRPs still remain challenging due to their NP-hard nature [3]. Recent studies on learning neural heuristics are gathering attention as promising extensions to traditional hand-crafted ones (e.g., [4–14]), where reinforcement learning (RL) [15] is usually exploited to train a deep neural network as an efficient solver without hand-crafted rules. A salient motivation is that deep neural networks may learn better heuristics by identifying useful patterns in an end-to-end and data-driven fashion.

Solutions to VRPs, i.e., routes, are sequences of nodes (customer and depot locations). Naturally, deep models for Natural Language Processing (NLP), which deal with sequence data as well, are ideal

---

*Zhiguang Cao and Wen Song are the corresponding authors.

35th Conference on Neural Information Processing Systems (NeurIPS 2021).

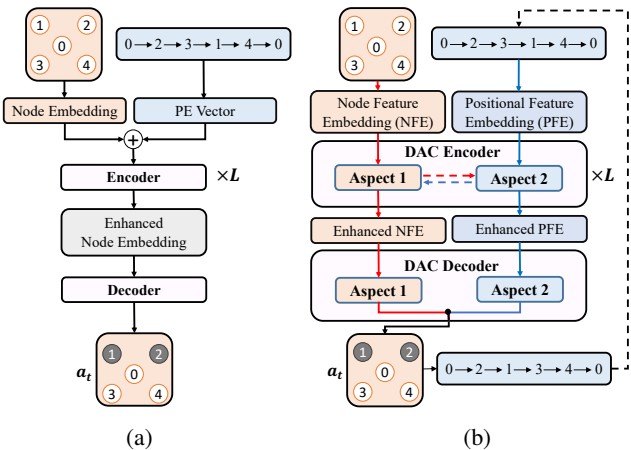

Figure 1: Transformer frameworks for VRPs. (a) Wu et al. [11] (the original one); (b) DACT (ours).

choices for encoding VRP solutions. Given its remarkable performance in NLP tasks, Transformer [16] is standing at the forefront in the learning based methods for VRPs (e.g., [5, 7, 8, 11–13, 17]). The original Transformer encodes a sentence, i.e., a sequence of words, into a unified set of embeddings by injecting word positional information into its word embeddings through positional encoding (PE). When it comes to VRPs, while is not required in *construction* models, positional information is critical for deep models that learn *improvement* heuristics since the input are solutions to be improved.

Although some success has been achieved, learning *improvement* heuristics for VRPs based on the original Transformer encoder is yet lacking from our perspective. Firstly, directly applying addition operation on PE vectors and the embeddings in absolute PE method (i.e., Figure 1(a)) could limit the representation of the model [18], as the mixed correlations[2] existing in the self-attention can bring unreasonable noises and random biases to the encoder (details in Appendix A). Secondly, existing PE methods tend to fuse the node and positional information into one unified representation. NLP tasks such as translation may benefit from this owing to the deterministic and instructive nature of the positional information. However, such design may not be optimal for routing tasks because the positional information therein can be non-deterministic and sometimes even random. This may cause disharmony or disturbance in the encoder and may thus deteriorate the performance. Finally, most VRPs seek the shortest loop of the nodes, making their solutions to be cyclic sequences. However, existing PE methods are only designated to encode linear sequences[3], which may fail to identify such circular input. As will be shown in our experiments, this could severely damage the generalization performance, since the cyclic feature of VRP solutions is not correctly reflected by the encoder.

In this paper, we address the above issues and contribute to the line of using RL to learn neural improvement heuristics for VRPs. We introduce the *Dual-Aspect Collaborative Transformer (DACT)*, where we revisit the solution representations and propose to learn separated groups of embeddings for the node and positional features of a VRP solution as shown in Figure 1(b). Our DACT follows the encoder-decoder structure. In the encoder, each set of embeddings encodes the solution mainly from its own aspect, and at the same time exploits a *cross-aspect referential attention mechanism* for better perceiving the consistence and differentiation with respect to the other aspect. The decoder then collects action distribution proposals from the two aspects and synthesizes them to output the final one. Meanwhile, we design a novel *cyclic positional encoding* (CPE) method to capture the circularity and symmetry of VRP solutions, which allows Transformer to encode cyclic inputs, and also boost the generalization performance for solving VRPs. As the last contribution, we design a simple yet effective curriculum learning strategy to improve the sample efficiency. This further leads to faster and more stable convergence of RL training. Extensive experiments show that our DACT can outperform existing Transformer based improvement models with fewer parameters, and also generalizes well across different sizes of synthetic and benchmark instances, respectively.

---

[2]The term *correlation* refers to the dot product between Query and Key in the self-attention module. The term *mixed correlation* refers to the case where Query and Key are projected from different types of embeddings.

[3]The relative PE method seems to help, however, it is found to be even worse than the absolute PE method for VRPs in Wu et al. [11], partly due to the disharmony issue caused by learning the unified representation.

## 2 Related work

### 2.1 Positional encoding (PE) in Transformer.

The original Transformer adopted the absolute PE method to describe the absolute position of elements in the sequence [16], especially for NLP. As formulated in Eq. (1), each generated positional embedding $p_i \in \mathbb{R}^d$ is added together with the $i$-th *word* embedding $x_i$ in the first layer of the encoder,

$$\alpha_{i,j}^{\text{Abs}} = \frac{1}{\sqrt{d}}((x_i + p_i)W^Q)((x_j + p_j)W^K)^T. \tag{1}$$

The relative PE method was further proposed in Shaw et al. [19] to better capture the relative order information. On the basis of absolute PE, it introduces an inductive bias to the attention as follows,

$$\alpha_{i,j}^{\text{Rel}} = \frac{1}{\sqrt{d}}((x_i + p_i)W^Q)((x_j + p_j)W^K + a_{j-i})^T, \tag{2}$$

where $a_{j-i} \in \mathbb{R}^d$ is learnable parameters for encoding the relative position $j-i$. To avoid the mixed and noisy correlations between word semantics and positional information in the above two PEs, the Transformer with United Positional Encoding (TUPE) [18] was proposed for NLP which utilizes separated projection metrics $W_x$ and $W_p$ for each information as follows,

$$\alpha_{i,j}^{\text{TUPE}} = \frac{1}{\sqrt{2d}}(x_i W_x^Q)(x_j W_x^K)^T + \frac{1}{\sqrt{2d}}(p_i W_p^Q)(p_j W_p^K)^T + b_{j-i}. \tag{3}$$

However, as mentioned previously, existing PE methods are less effective for VRPs since they simply fuse the node and positional information into one unified set of embeddings during or after the calculation of the attention correlation $\alpha_{i,j}$. Meanwhile, they are also unable to properly encode and handle cyclic input sequences as in VRP solutions.

### 2.2 Deep models for VRP.

Various deep architectures such as Recurrent Neural Network (RNN), Graph Neural Network (GNN), and Transformer have been employed in solving VRPs.

**RNN based models.** As the pioneering work of neural VRP solvers, Pointer Network adopted RNN and supervised learning to solve TSP [20] (extended to RL in Bello et al. [21] and CVRP in Nazari et al. [22]). While the models in [20–23] learn *construction* heuristics, NeuRewriter [4] learns *improvement* heuristic for CVRP using LSTM to encode the positional information of a solution. In Hottung et al. [13], the conditional variational autoencoder was adopted to learn a continuous and latent search space for VRP, where high-quality solutions were taken as input and encoded by RNNs. However, recurrence structures in RNN are less efficient in both representation and computation [5].

**GNN based models.** In Dai et al. [24], GNN was combined with Q-learning for solving TSP. Based on supervised learning, Joshi et al. [6] used GNN to learn heatmaps that prescribe the probability of each edge appearing in the optimal TSP tour. This idea was extended in Fu et al. [25] with additional components such as graph sampling and heatmap merging to enable generalization to larger TSP instances. These models often require post-processing to construct feasible solutions from heatmaps (e.g., beam search [6], Monte-Carlo tree search [25], and dynamic programming [26]).

**Transformer based models.** The Attention Model (AM) by Kool et al. [5] was recognized as the first success of Transformer based models for VRPs. Based on AM, Xin et al. [7] proposed a Multi-Decoder AM that learns multiple diverse policies for better performance. In Kwon et al. [8], the RL algorithm of AM was improved which leaded to a new solver, i.e., POMO (Policy Optimization with Multiple Optima), and achieved the state-of-the-art performance. However, POMO is still lacking in generalization. Besides these *construction* models, Transformer was also explored to learn *improvement* heuristics. Hottung and Tierney [27] learned first neural large neighborhood search algorithm for VRPs. Lu et al. [12] proposed the L2I model that learns to select local search operators from a pool of traditional ones. Both methods used a Transformer-style encoder, but the positional information is captured in the node features (information of previous and next nodes) instead of using PE methods. Though L2I was shown to outperform LKH3 [28], it is limited to CVRP and the required time is considerably long. Wu et al. [11] proposed a Transformer model which learns to pick node pair in each step to perform a pairwise local operator (e.g., 2-opt). However, it suffers from the inaccurate representation of positional information given the original Transformer encoder.

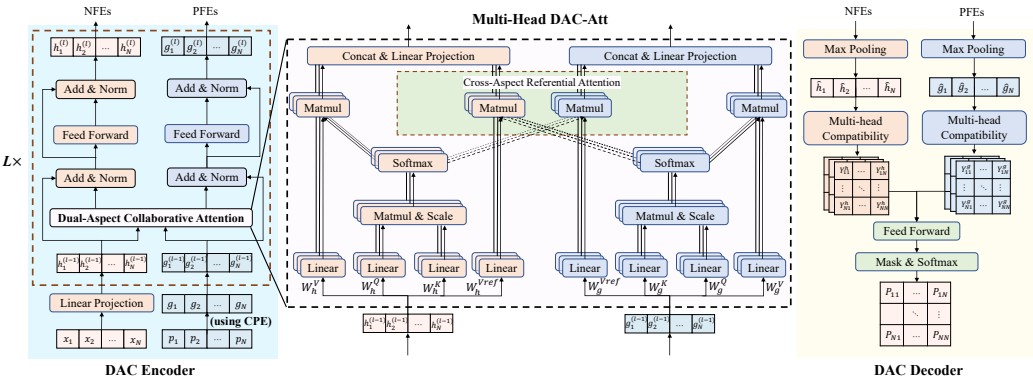

Figure 3: Architecture of our policy network, dual-aspect collaborative Transformer (DACT).

## 3 Problem formulation

We define a VRP instance as a group of $N$ nodes to visit, where the *node feature* $x_i$ of node $i$ contains 2-dim coordinates and other problem-specific features (e.g., customer demand). A solution $\delta$ consists of a sequence of nodes visited in order where we denote $p_i$ to be the position (indices) of node $i$ in the solution which is deemed as the *positional feature* of node $i$. The objective is to minimize the total travel distance $D(\delta)$ under certain problem-specific constraints.

Starting with an initial yet complete solution, our neural RL policy tries to improve the solution iteratively. At each step, the policy automatically selects a pair of nodes and locally adjusts the solution using a preset pairwise operator such as *2-opt*, *insert*, or *swap*. As illustrated in Figure 2, given a node pair $(i, j)$, the *2-opt* operator adjusts a solution by reversing the segment between node $i$ and node $j$; the *insert* operator adjusts a solution by placing node $i$ after node $j$; and the *swap* operator

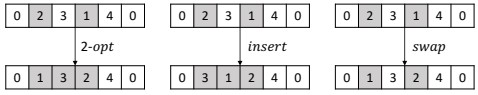

Figure 2: Illustration examples of three pairwise operators for routing problems when node pair $(i = 2, j = 1)$ is specified for operating. From left to right: *2-opt*, *insert*, and *swap*.

adjusts a solution by exchanging the position of node $i$ and node $j$. Such operation is repeated until reaching the step limit $T$ and we model it in the form of Markov Decision Process (MDP) as follows.

**State.** For an instance with $N$ nodes, a state describes current solution $\delta_t$ using its node and positional features of each node, i.e., $s_t = \Psi(\delta_t) = \{x_1^t, ..., x_N^t, p_1^t, ..., p_N^t\}$.

**Action.** The action $a_t = (i, j)$ specifies a node pair $(i,j)$ for the pairwise operator.

**Reward.** The reward function is defined as, $r_t = D(\delta_t^*) - min\left[D(\delta_{t+1}), D(\delta_t^*)\right]$ where $\delta_t^*$ is the best incumbent solution found until time $t$. It refers to the immediate reduced cost at each step with respects to the best incumbent solution, which ensures the cumulative reward equal to the total reduced cost over the initial solution. Hence the reward $r_t > 0$ if and only if a better solution is found.

**Policy.** The policy $\pi_\theta$ is parameterized by the proposed DACT model with parameters $\theta$. At each time step, the action $(i, j)$ is obtained by sampling the stochastic policy for both training and inference.

**Transition.** The next state $s_{t+1}$ is originated from $s_t$ by performing the preset pairwise operator on the given node pair (action). Our state transient is deterministic, in the sense that it always accepts the next solution as the next state (infeasible solutions will be masked), regardless of its objective value. With such simple rule, the RL agent is expected to automatically learn how to combine multiple steps of simple local movements to achieve better solutions, even if some of them may worsen the current solution. Note that the step limit $T$ can be any user-specified value according to the allowed time budget. Hence, our MDP can have infinite horizon and we consider the reward discount factor $\gamma < 1$.

## 4 Dual-aspect collaborative Transformer model

We now present the details of our *Dual-Aspect Collaborative Transformer* (DACT). The concrete architecture of DACT is presented in Figure 3, where we take the TSP with $N$ nodes as an illustration

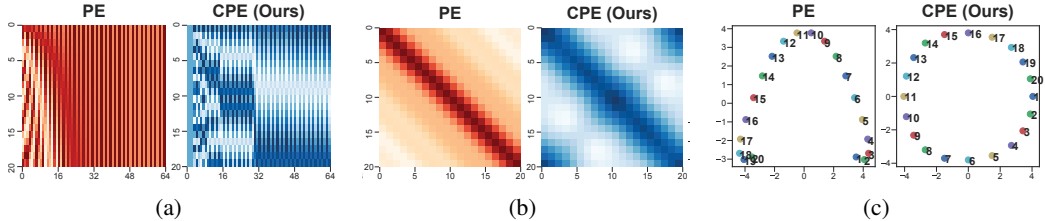

(a)  (b)  (c)

Figure 5: Comparison of our CPE method with absolute PE method on a TSP instance with 20 nodes. (a) the embedding vectors, (b) the correlations (dot products) between every two embeddings, and (c) the top two principal components after PCA (principal component analysis) projection.

example. Our DACT leverages separate *aspects* of embeddings to encode a VRP solution. In the DAC *encoder*, the self-attention correlations are computed individually for each aspect, and a *cross-aspect referential attention mechanism* is proposed to enable one aspect to effectively exploit attention correlations from the other aspect as optional references. The DAC *decoder* then collects action distribution proposals from both aspects and synthesize them to the final one.

## 4.1 Dual-aspect solution representation

Specifically, we propose to learn two sets of embeddings, i.e., the *node feature embeddings* (NFEs) for node representation and the *positional feature embeddings* (PFEs) for positional representation.

**NFEs.** Following [5, 11], the NFE $h_i$ of node $i$ is initialized as the linear projection of its node feature $x_i$ with output dimension[4] $dim = 64$.

**PFEs.** The PFE $g_i$ of the positional feature $p_i$ is initialized as a real-valued vector ($dim = 64$) by applying our cyclic positional encoding (CPE), which is designed based on cyclic Gray codes [29].

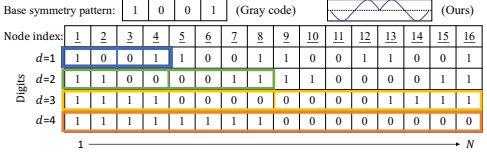

As illustrated in Figure 4, the cyclic Gray codes present a cyclic property ('1110' in the last column is adjacent to '1111' in the first column) and an adjacency similarity property (any codes in adjacent columns only differ in one digit), both of which are desirable for cyclic sequences. To preserve these properties in designing our CPE, we follow two observed patterns: 1) each numerical digit contains a periodic cycle with reflectional symmetry, e.g., the '10|01' in the lowest digit; and 2) the higher the numerical digit, the longer the period. Accordingly, we create similar patterns based on the sinusoidal functions in Eq. (4), where a periodic function with period $\frac{4\pi}{\omega_d}$ (induced by modulus) is used to generate one base symmetry pattern (the top right in Figure 4),

Figure 4: An example of cyclic Gray code where 4 digits are used to encode $N = 16$ nodes. The top left shows the base symmetry pattern '1001' in Gray code, and the top right plots its representation in our method.

$$\overrightarrow{g_i}^{(d)} := \begin{cases} sin(\omega_d \cdot (z(i) \bmod \frac{4\pi}{\omega_d}) - \frac{2\pi}{\omega_d} \ ), & \text{if } d \text{ is even} \\ cos(\omega_d \cdot (z(i) \bmod \frac{4\pi}{\omega_d}) - \frac{2\pi}{\omega_d} \ ), & \text{if } d \text{ is odd} \end{cases} \tag{4}$$

where $z(i) = \frac{i-1}{N} \frac{2\pi}{\omega_d} \left\lceil \frac{N+1}{2\pi/\omega_d} \right\rceil$ is to make $N$ nodes linearly spaced in the generated pattern; the angular frequency $\omega_d$ is decreasing along the dimension to make the wavelength longer within the range $\left[ N^{\frac{1}{\lfloor dim/2 \rfloor}}, N \right]$ (see Appendix B for details). In Figure 5, we visualize the comparison between the absolute PE and our CPE for encoding a TSP instance of 20 nodes. Figure 5(a) demonstrates that our real-valued base symmetry pattern has a longer cyclic period as the digit grows. Figure 5(b) indicates that our method (blue) is able to correctly reflect the adjacency between the head and tail of the cyclic sequence whereas the PE method (red) fails to do so. Figure 5(c) verifies that our CPE vectors are well distributed in space with desired cyclic and adjacency similarity properties.

---

[4]Different from Wu et al. [11], we reduce the dimension of the embeddings from 128 to 64.

## 4.2 The encoder

The encoder consists of $L = 3$ stacked DAC encoders. In each DAC encoder, we retain relatively independent encoding stream for NFEs and PFEs as in Eq. (5) and Eq. (6), respectively, each of which consists of a shared Dual-Aspect Collaborative Attention (**DAC-Att**) sub-layer and an independent feed-forward network (**FFN**) sub-layer. DAC-Att takes both sets of embeddings as input and then outputs their respective enhanced embeddings, i.e., NFEs $\{\tilde{\boldsymbol{h}}\}_{i=1}^N$ and PFEs $\{\tilde{\boldsymbol{g}}\}_{i=1}^N$. Each sub-layer is followed by skip connection [30] and layer normalization [31] as same as the original Transformer.

$$h_i^{(l)} = \mathbf{LN}\left(h_i' + \mathbf{FFN}_h^{(l)}(h_i')\right), \, h_i' = \mathbf{LN}\left(h_i^{(l-1)} + \tilde{\boldsymbol{h}}_i^{(l)}\right), \tag{5}$$

$$g_i^{(l)} = \mathbf{LN}\left(g_i' + \mathbf{FFN}_g^{(l)}(g_i')\right), \, g_i' = \mathbf{LN}\left(g_i^{(l-1)} + \tilde{\boldsymbol{g}}_i^{(l)}\right). \tag{6}$$

**DAC-Att.** The DAC-Att sub-layer enhances each set of embedding from its own aspect, while leveraging attention correlations from the other aspect to achieve the synergy. Given the two sets of embeddings[5], $\{h_i\}_{i=1}^N$ and $\{g_i\}_{i=1}^N$, we first compute the self-attention correlation from both aspects,

$$\alpha_{i,j}^h = \frac{1}{\sqrt{d_k}}\left(h_i W_h^Q\right)\left(h_j W_h^K\right)^T, \quad \alpha_{i,j}^g = \frac{1}{\sqrt{d_k}}\left(g_i W_g^Q\right)\left(g_j W_g^K\right)^T, \tag{7}$$

where independent matrices $W_h^Q, W_h^K, W_g^Q$ and $W_g^K \in \mathbb{R}^{dim \times d_k}$ are used to calculate *queries* and *keys*. The obtained correlations are further normalized to $\tilde{\alpha}_{i,j}^h$ and $\tilde{\alpha}_{i,j}^g$ via Softmax. Note that the correlations are computed from their own aspect, which eliminates possible noises and conduces to correctly describe the incompatible node pair relationships in different aspects of VRP solutions.

We then exploit a *cross-aspect referential attention* mechanism, which allows computed correlations to be shared between each other, as additional references for both contradistinction and collaboration,

$$\mathrm{out}_i^h = \mathrm{Concat}\left[\sum_{j=1}^N \tilde{\alpha}_{i,j}^h\left(h_j W_h^V\right), \sum_{j=1}^N \tilde{\alpha}_{i,j}^g\left(h_j W_h^{Vref}\right)\right], \tag{8}$$

$$\mathrm{out}_i^g = \mathrm{Concat}\left[\sum_{j=1}^N \tilde{\alpha}_{i,j}^g\left(g_j W_g^V\right), \sum_{j=1}^N \tilde{\alpha}_{i,j}^h\left(g_j W_g^{Vref}\right)\right], \tag{9}$$

where $W_h^V, W_g^V \in \mathbb{R}^{dim \times d_v}$ are trainable parameter matrices for formulating *values* in each aspect; and $W_h^{Vref}, W_g^{Vref} \in \mathbb{R}^{dim \times d_v}$ are parameter matrices for each aspect to generate *referential values*. We finally use the *multi-head attention* to get NFEs $\tilde{h}_i$ and PFEs $\tilde{g}_i$ as follows,

$$\tilde{h}_i, \tilde{g}_i = \mathbf{DAC\text{-}Att}\left(W^Q, \, W^K, W^V, W^{V_{ref}}, W^O\right),$$
$$\tilde{h}_i = \mathrm{Concat}\left[\mathrm{head}_{i,1}^h, ..., \mathrm{head}_{i,m}^h\right] W_h^O, \, \tilde{g}_i = \mathrm{Concat}\left[\mathrm{head}_{i,1}^g, ..., \mathrm{head}_{i,m}^g\right] W_g^O, \tag{10}$$

where $\mathrm{head}_{i,k}^h = out_{i,k}^h$, $\mathrm{head}_{i,k}^g = out_{i,k}^g$, and $W_h^O, W_g^O \in \mathbb{R}^{2md_v \times dim}$ are trainable parameter matrices. In our model, we adopt $m = 4$ and $d_k = d_v = 16$.

**FFN.** Our FFN sub-layer has only one hidden layer with 64 hidden unites and adopts the ReLU activation function. The parameters of $\mathbf{FFN}_h$ and $\mathbf{FFN}_g$ are different for each group of embeddings.

## 4.3 The decoder

In the DAC decoder, the two sets of embeddings $\{h_i^{(L)}\}_{i=1}^N$ and $\{g_i^{(L)}\}_{i=1}^N$ are first passed through a **Max-pooling** sub-layer and a multi-head compatibility (**MHC**) sub-layer to independently generate diversified node-pair selection proposals from their own aspect, which are then aggregated through a feed-forward aggregation (**FFA**) sub-layer for output.

---

[5]We omit the encoder index $l$ for better readability.

**Max-pooling**. For each set of embeddings, we adopt the max-pooling sub-layer in Wu et al. [11] to aggregate the global representation of all $N$ embeddings into each respective one[6].

**MHC**. The compatibility sub-layer computes the attention correlations for each embedding pair, where the obtained correlations with size $N \times N$ will be deemed as a proposal distribution for node pair selection. Our correlations are computed based on multiple heads for diversity. And we calculate separated attention score matrices $Y_k^h, Y_k^g \in \mathbb{R}^{N \times N}$ (of head $k$) from the two aspects independently. Accordingly, the action distribution proposals would be different due to their aspect-specific focus and cognitions of the current solution, which will provide the subsequent FFA layer with a rich pool of proposals and allow our model to be more flexible and robust.

**FFA**. Once all proposals from two aspects are collected, a FFN with four layers (dimensions are $2m$, 32, 32 and 1, respectively) and ReLU activation is used to aggregate them,

$$\tilde{Y}_{i,j} = \mathbf{FFA}\left(Y_{i,j,1}^g, ..., Y_{i,j,m}^g, Y_{i,j,1}^h, ...Y_{i,j,m}^h\right), \tag{11}$$

where $m = 4$ is the number of heads; and the output $\tilde{Y}_{i,j}$ is a scalar indicating the likelihood of selecting node pair $(i, j)$ as an action. Afterwards, we apply $\hat{Y}_{ij} = C \cdot \mathrm{Tanh}(\tilde{Y}_{i,j})$ with $C = 6$ to control the entropy, and mask [7] the infeasible node pairs $(i', j')$ as $\hat{Y}_{i'j'} = -\infty$. Lastly, the likelihoods are normalized using Softmax function to obtain the final action distribution $P_{i,j}$.

## 4.4 Reinforcement learning algorithm

We adopt the proximal policy optimization [32] with $n$-step return estimation for training (details are given in Appendix C), and design a *curriculum learning* (CL) strategy for better sample efficiency.

**Curriculum learning strategy.** The strategy in Wu et al. [11] sets a maximum of $T_{train}$ steps for training and estimates future returns by *bootstrapping* [33]. However, due to the concern of training cost, $T_{train}$ is usually much smaller than actual $T$ for inference (e.g., 200 v.s. 10k), which may leave the agent a poor chance of observing high-quality solutions (states) during training. Consequently, it may cause high variance for bootstrapping because the value function is mostly fitted on low-quality solutions and may render it less knowledgeable in estimating long-term future returns accurately. In this paper, we tackle this issue by a simple yet efficient strategy which gradually prescribes higher-quality solutions as the initial states for training. In doing so, 1) it increases the probability for the agent to observe better solutions and thus reduce the variance of the value function; 2) it increases the difficulty of the learning task (higher-quality solutions are harder to improve) in a gradual manner and achieves better sample efficiency [34]. In practice, those higher-quality solutions can be easily achieved by improving the randomly generated ones using the current policy for a few $T_{init}$ steps, where $T_{init}$ could be slightly increased as the epoch grows.

## 5 Experiments

We evaluate our DACT model on two representative routing problems, i.e., TSP and CVRP [5, 8, 11]. For each problem, we abide by existing conventions to randomly generate instances on the fly for three sizes, i.e., $N = 20$, 50 and 100. Initial experiments with three operators including *2-opt*, *swap* and *insert* show that *2-opt* performs best for both TSP and CVRP (with *insert* better than *swap*), hence we report results of our method based on *2-opt*. Following [4, 11, 27] we use randomly generated initial solutions for training and the solutions generated by the greedy algorithm for inference. Since each problem has its own constraints and node features, we adjust the input, feasibility masks, and problem-dependent hyperparameters for each problem, the details of which are provided in Appendix D and E. The DACT is trained and tested on a server equipped with TITAN RTX GPU cards and Intel i9-10940X CPU at 3.30 GHz. Our code in PyTorch are available here[8].

---

[6]E.g., for $\hat{h}$, $\hat{h}_i = h_i^{(L)} W_h^{Local} + \max\left[\{h_i^{(L)}\}_{i=1}^N\right] W_h^{Global}$, where $W_h^{Local}, W_h^{Global} \in \mathbb{R}^{64 \times 64}$ are parameters.

[7]Besides, we also mask all the diagonal elements since they are not meaningful to the pair-wise operators, and the node pair selected at the last step to forbid possible dead loops [11].

[8]https://github.com/yining043/VRP-DACT

Table 1: Comparison with various baselines on TSP and CVRP.

| Method | N=20 | | | N=50 | | | N=100 | | |
|---|---|---|---|---|---|---|---|---|---|
| | Obj. | Gap | Time | Obj. | Gap | Time | Obj. | Gap | Time |
| **TSP** | | | | | | | | | |
| Concorde | 3.83 | - | (3m) | 5.70 | - | (10m) | 7.76 | - | (1h) |
| LKH | 3.83 | 0.00% | (38s) | 5.70 | 0.00% | (5m) | 7.76 | 0.00% | (20m) |
| OR-Tools | 3.86 | 0.94% | (42s) | 5.85 | 2.87% | (5m) | 8.06 | 3.86% | (23m) |
| Neural-2-Opt [23] | 3.84[‡] | **0.00%** | (15m) | 5.70 | 0.12% | (29m) | 7.83 | 0.87% | (41m) |
| Wu et al. [11] (T=5k) | 3.83 | **0.00%** | (1h) | 5.70[‡] | 0.20% | (1.5h) | 7.87 | 1.42% | (2h) |
| DACT (T=1k) | 3.83 | 0.04% | {7s}(24s) | 5.70 | 0.14% | {16s}(1m) | 7.89 | 1.62% | {48s}(4m) |
| DACT (T=5k) | 3.83 | **0.00%** | {32s}(2m) | 5.70 | 0.02% | {2m}(6m) | 7.81 | 0.61% | {4m}(18m) |
| DACT (T=10k) | 3.83 | **0.00%** | {1m}(5m) | 5.70 | 0.01% | {3m}(13m) | 7.79 | 0.37% | {8m}(40m) |
| DACT×4 augment | 3.83 | **0.00%** | {3m}(10m) | 5.70 | **0.00%** | {10m}(1h) | 7.77 | **0.09%** | {29m}(2.5h) |
| GCN-BS [6] | 3.84[‡] | 0.01% | (12m) | 5.70 | 0.01% | (18m) | 7.87 | 1.39% | (40m) |
| AM-sampling [5] | 3.84[‡] | 0.08% | (5m) | 5.73 | 0.52% | (24m) | 7.94 | 2.26% | (1h) |
| MDAM-BS [7] | 3.84[‡] | **0.00%** | (3m) | 5.70 | 0.03% | (14m) | 7.79 | 0.38% | (44m) |
| POMO [8] | 3.83 | 0.04% | (1s) | 5.70[‡] | 0.21% | (2s) | 7.80 | 0.46% | (11s) |
| POMO×8 augment [8] | 3.83 | **0.00%** | (3s) | 5.69[‡] | 0.03% | (16s) | 7.78 | 0.15% | (1m) |
| DPDP (100k) [26] | - | - | - | - | - | - | 7.77[‡] | 0.00% | (3h) |
| CVAE-Opt-DE [13] | - | **0.00%**[#] | 11m[#] | - | 0.02%[#] | 22m[#] | - | 0.34%[#] | 55m[#] |
| **CVRP** | | | | | | | | | |
| LKH | 6.14 | 0.00% | 1h | 10.38 | 0.00% | 4h | 15.68 | 0.00% | 8h |
| OR-Tools | 6.46 | 5.68% | 2m | 11.27 | 8.61% | 13m | 17.12 | 9.54% | 46m |
| NeuRewriter [4] | 6.15[#] | - | 6m[#] | 10.51[#] | - | 11m[#] | 16.10[#] | - | 17m[#] |
| NLNS [27] | 6.19[#] | - | 6m[#] | 10.54[#] | - | 11m[#] | 15.99[#] | - | 16m[#] |
| Wu et al. [11] (T=5k) | 6.12[‡] | 0.39% | (2h) | 10.45 | 0.70% | (4h) | 16.03[‡] | 2.47% | (5h) |
| DACT (T=1k) | 6.15 | 0.28% | {16s}(33s) | 10.61 | 2.13% | {43s}(2m) | 16.17 | 3.18% | {2m}(5m) |
| DACT (T=5k) | 6.13 | -0.00% | {1m}(3m) | 10.48 | 1.01% | {3m}(8m) | 15.92 | 1.55% | {8m}(23m) |
| DACT (T=10k) | 6.13 | -0.04% | {2m}(6m) | 10.46 | 0.79% | {6m}(16m) | 15.85 | 1.12% | {16m}(45m) |
| DACT×6 augment | 6.13 | **-0.08%** | {11m}(35m) | 10.39 | **0.14%** | {32m}(1.5h) | 15.71 | **0.19%** | {1.5h}(4.5h) |
| AM-sampling [5] | 6.25 | 1.87% | (6m) | 10.62 | 2.40% | (28m) | 16.23[‡] | 3.72% | (2h) |
| MDAM-BS [7] | 6.14 | 0.18% | (5m) | 10.48 | 0.98% | (15m) | 15.99[‡] | 2.23% | (1h) |
| POMO [8] | 6.17[‡] | 0.82% | (1s) | 10.49 | 1.14% | (4s) | 15.83 | 0.98% | (19s) |
| POMO×8 augment [8] | 6.14[‡] | 0.21% | (5s) | 10.42 | 0.45% | (26s) | 15.73 | 0.32% | (2m) |
| DPDP (100k) [26] | - | - | - | - | - | - | 15.69[‡] | 0.31% | (6h) |
| CVAE-Opt-DE [13] | 6.14[#] | - | 21m[#] | 10.40[#] | - | 41m[#] | 15.75[#] | - | 1.5h[#] |

[#] the obj. values, gaps or time are obtained based on 2,000 instances in their original papers, and not directly comparable to ours.
[‡] the obj. values obtained by Concorde or LKH may be slightly different from ours since the 10,000 instances are randomly generated. E.g., for TSP50, the optimal values according to our running of Concorde is 5.70, while 5.69 in POMO and Wu et al.. We thus focus more on gaps.

## 5.1 Comparison studies

In Table 1, we compare our DACT with, (1) learning based *improvement* methods, including Wu et al. [11], Neural-2-Opt [23] (TSP only), NeuRewriter [4] (CVRP only), NLNS [27] (CVRP only), (2) learning based *construction* methods, including AM-sampling [5], GCN-BS [6] (TSP only), MDAM-BS [7], POMO [8], (3) conventional optimization algorithms equipped with learning based component(s), including DPDP [26], CVAE-Opt-DE [13], and (4) strong conventional solvers including Concorde [35], LKH [28, 36], and OR-Tools [37]. Though L2I [12] can outstrip LKH on CVRP, we do not inlude it as a baseline since it requires a prohibitively longer inference time than others[9]. All results are averaged over 10,000 randomly generated instances unless specified otherwise (e.g., the ones marked with # only infer 2,000 instances), and we report the metrics of objective values, (optimality) gaps and run time. Regarding baselines, we follow the results reported in their original papers, which may not include all the three metrics. For TSP, Concorde is adopted to get the optimal solutions, based on which the optimality gaps of other methods are calculated. CVRP is harder to be solved optimally, and the gaps are calculated based on solutions of LKH. Note that even for the baselines which infer 10,000 random instances, their objective values might be slightly different from ours (e.g., the ones marked with ‡), therefore we focus more on gaps for fair comparison. The run time is also hard to compare due to various factors (e.g., GPU/CPU models, batch sizes, Python v.s. C++). For DACT, we report the time for inferring all 10,000 instances with multiple GPU cards in "()", and a small batch (512 instances) with one single GPU card in "{}".

Pertaining to TSP, our DACT with inference step limit of 5,000 (T=5k) outperforms the traditional solver OR-Tools and all improvement models in terms of optimality gap, including Wu et al. [11] which directly adopted the original Transformer encoder. It also outstrips construction methods

---

[9]L2I needs 167 days for 10,000 CVRP100 instances, as estimated from 24min/instance in its original paper.

Table 2: Generalization performance. (a) DACT v.s. baselines on benchmark datasets (up to 200 customers, see Appendix E.4 for detailed results and discussion); (b) PE v.s. CPE on different sizes.

| Method | TSPLIB | CVRPLIB |
|---|---|---|
| OR-Tools [37] | 3.34% | 8.06% |
| AM-sampling [5] | 22.83% | 26.66% |
| POMO [8] | 10.06% | 6.10% |
| Wu et al. [11] | 4.17% | 5.20% |
| DACT | **2.07%** | **3.41%** |

(a)

| Method | N=20 | | N=100 | |
|---|---|---|---|---|
| | Obj. | Gap | Obj. | Gap |
| DACT-PE (T=5k) | 3.84 | 0.21% | 8.38 | 7.93% |
| DACT-CPE (T=5k) | **3.83** | **0.10%** | **7.99** | **2.98%** |
| Wu et al. [11] (T=5k) | 3.91 | 2.14% | 9.03 | 16.37% |
| OR-Tools [37] | 3.83 | 0.00% | 8.06 | 3.87% |

(b)

Table 3: Dual v.s. single aspect representation

| Steps | Method | # Params | N=50 | N=100 |
|---|---|---|---|---|
| T=1k | SA-T | 0.37M | 0.35% (1m) | 3.49% (3m) |
| | DACT | 0.29M | **0.14%** (1m) | **1.62%** (4m) |
| T=5k | SA-T | 0.37M | 0.05% (5m) | 1.55% (16m) |
| | DACT | 0.29M | **0.02%** (6m) | **0.61%** (18m) |

including AM-sampling and GCN-BS on TSP100. With larger step limit T=10k, our DACT further boosts the solution qualities and outperforms other construction methods including MDAM-BS (beam search), and POMO (the current state-of-the-art). To further reduce the gaps, we also leverage the data augmentation technique in POMO (which considers flipping node coordinates without changing the optimal solution) to solve same instances multiple times in different ways. Although the inference time increases (we run data augmentation in serial on the same GPUs), our DACT with 4 augments not only outstrips POMO with 8 augments but also achieves the lowest objective values and gaps among all purely learning based models. In particular, our method almost optimally solved TSP20 and TSP50 with gap lower than 0.005%, and 0.09% on TSP100, which is superior to most of the recent neural solvers. Pertaining to CVRP, our DACT with T=5k produces lower gaps than that of improvement models including NeuRewriter and NLNS. It also performs much better than Wu et al. [11] except on CVRP50. With T=10k and 6 augments[10], our DACT exhibits even better performance than the highly specialized heuristic solver LKH on CVRP20 and delivers the smallest gap of 0.19% on CVRP100 against other neural solvers including POMO with 8 augments. Besides, our DACT is also competitive to DPDP which leverages learnt heatmap and dynamic programming to search solutions. Though DPDP (100k) can solve TSP100 instances almost optimally, our DACT is more efficient than DPDP on CVRP100. Compared with CVAE-Opt-DE, despite that it is averaged over fewer instances and integrated with differential evolution, our objective values are still lower.

In terms of the inference time, our DACT is highly competitive against all neural solvers except POMO which learns a *construction* model by sampling diverse trajectories. However, when it comes to the generalization performance on benchmark datasets, i.e., TSPLIB [38] and CVRPLIB [39] in Table 2(a), DACT produces significantly lower average gaps than the POMO with 8 augments, which indicates that our DACT is more advantageous in practice despite its longer inference time. On the other hand, it is possible to adopt a similar diverse rollout strategy for DACT to find better solutions earlier, or explore other model compression techniques such as the knowledge distillation [40] to learn a lighter DACT model for faster inference. Since our focus is to ameliorate Transformer for neural improvement solvers, we will investigate these possibilities in the future.

## 5.2 Ablation studies

**Dual-aspect representation.** In Table 3, we evaluate the effectiveness of our dual-aspect representation against the single-aspect one (SA-T) on TSP50 and TSP100, where SA-T mainly follows the Transformer in Wu et al. [11] but equipped with the CPE, multi-head attentions and CL strategy for fair comparison. We observe that our DACT with fewer parameters consistently outperforms SA-T, which verifies the effectiveness of the dual-aspect representation.

---

[10]The gap will further decrease to 0.09% if we adopt 8 augments, with run time of about 7h.

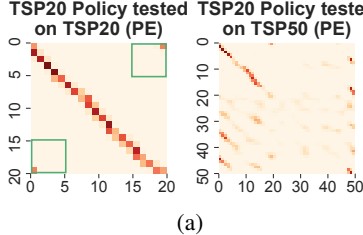 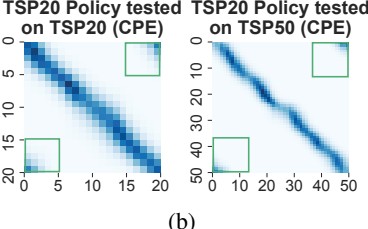

(a)                                      (b)

Figure 6: Visualization of the attention scores for the encoder when a trained model is used to solve instances with a larger size. (a) using PE method; (b) using CPE method (ours).

**Cyclic positional encoding.** Here we show that CPE significantly improves the generalization performance across different problem sizes. In Table 2(b), we record the results of our DACT with PE and CPE, and Wu et al. [11], when the model trained on TSP50 is directly used to solve instances from TSP20 and TSP100 with T=5k. We see that even with PE, our DACT outperforms Wu et al. [11]. Further equipped with CPE, DACT outstrips DACT-PE and OR-Tools on TSP100. We continue to compare the two DACT variants by visualizing their attention scores. As depicted in Figure 6(a), although the absolute PE is designed for linear sequences, it did attempt to capture the circularity of

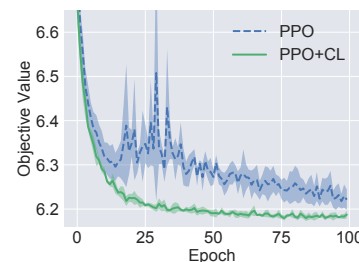

Figure 7: Training curves of PPO with and without CL on CVRP20 (random seeds 1-5).

VRP solutions (as highlighted in the green boxes) after training. However, the ability to perceive such properties significantly drops when generalizing over different problem size, which instead engenders random attention scores when generalizing to larger size (see right side of Figure 6(a)). In contrast, our DACT with CPE is able to capture the circularity as depicted in Figure 6(b), which verifies the effectiveness of CPE in representing cyclic sequences (i.e., VRP solutions).

**Curriculum learning (CL) strategy.** In Figure 7, we plot the training curves of PPO algorithm with and without our CL strategy, where the results are averaged over 5 independent runs with 90% confidence intervals. It shows that our CL strategy significantly improves the sample efficiency while reducing the variance of training, which aligns with our analysis in Section 4.4.

## 6 Conclusions and future work

In this paper, we present a novel DACT model for routing problems. It learns separate groups of embeddings for the node and positional features, and is equipped with cyclic positional encoding (CPE) to capture the circularity and symmetry of VRP solutions. A curriculum learning (CL) strategy is also exploited to improve the RL training efficiency. Extensive experiments on both synthetic and benchmark datasets justified the effectiveness of DACT in terms of both inference and generalization. A potential limitation is that DACT is more useful for learning improvement models at present. In the future, we will investigate how to extend DACT to construction models, and how to speed up the DACT through diverse rollouts or model compression techniques. It is also interesting to apply the proposed CPE to develop Transformer based model for other tasks where the cyclic property is also important, e.g., encoding circular DNA/RNA structures in computational biology [41, 42].

## Acknowledgments and Disclosure of Funding

This work was supported in part by the National Natural Science Foundation of China under Grant 61803104 and Grant 62102228, in part by the Young Scholar Future Plan of Shandong University under Grant 62420089964188, and in part by the A*STAR CyberPhysical Production System (CPPS) - Towards Contextual and Intelligent Response Research Program, under the RIE2020 IAF-PP Grant A19C1a0018, and Model Factory@SIMTech.

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
