# Learning to Iteratively Solve Routing Problems with Dual-Aspect Collaborative Transformer (Appendix)

## A  Issues in existing Transformer-based model for VRP

We investigate the issues of mixed correlations and noisy biases when the absolute PE method of Transformer is directly used to learn *improvement* heuristics in Wu et al. [11]. By fusing the node feature embedding $h_i$ and the positional feature embedding $g_i$ through an addition operator, four attention query terms from input $i$ to $j$ exist during the self-attention of the encoder as follows,

$$
\begin{aligned}
\alpha_{i,j}^{Abs} &= \frac{1}{\sqrt{d_k}} \left( (h_i + g_i)W^Q \right) \left( (h_j + g_j)W^K \right)^T \\
&= \frac{1}{\sqrt{d_k}} \left( h_i W^Q \right) \left( h_j W^K \right)^T + \frac{1}{\sqrt{d_k}} \left( g_i W^Q \right) \left( g_j W^K \right)^T \\
&+ \frac{1}{\sqrt{d_k}} \left( h_i W^Q \right) \left( g_j W^K \right)^T + \frac{1}{\sqrt{d_k}} \left( g_i W^Q \right) \left( h_j W^K \right)^T,
\end{aligned}
\tag{12}
$$

where we call them *node-to-node*, *position-to-position*, *node-to-position*, and *position-to-node*, respectively. Obviously, they all share the same projection matrices $W^Q$ and $W^K$, which might be unreasonable since they are used to represent correlations of different information [18]. Furthermore, the last two terms are essentially computing the mixed correlations across different information. Intuitively, queries from the location of a node (node feature) to the index of another node (positional feature) would be meaningless and vice versa. Such design may further bring noisy biases to routing problems. To verify this, we visualize the above four attention terms using a pre-trained model of Wu et al. [11] on a sampled batch of instances for TSP20. As shown in Figure 8, the last two correlations (node-to-position and position-to-node) seem to unreasonably present some random patterns across different node pairs, e.g., all nodes tend to have strong correlations with the ones appeared close to the end of the solution. This may yield biased attention, and thus affect the accuracy and the performance of the learned heuristics. In contrast, our DACT avoids such mixed correlations by learning feature embeddings for two aspects separately without fusing them into a unified representation.

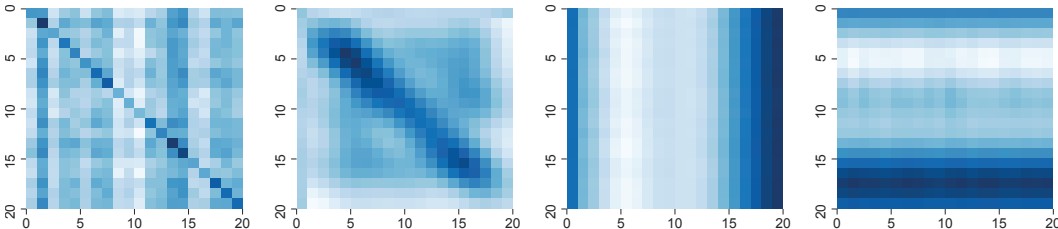

Figure 8: Visualizations of correlations on a trained model of Wu et al. [11]. From left to right: correlation for node-to-node, position-to-position, node-to-position, and position-to-node, respectively.

## B  Details of CPE

With the angular frequencies $\omega_d$ decreasing along the vector dimension to make the wavelength longer (the wavelength is chosen from $[N^{\frac{1}{\lfloor dim/2 \rfloor}}, N]$), we adopt Eq. (13) to determine the specific frequencies. Empirically, we fix the last half of the angular frequencies to $N$ (the largest value) to better preserve the desired cyclic and adjacency similarity properties.

$$
\omega_d = \begin{cases} \frac{3\lfloor d/3 \rfloor + 1}{dim} \left( N - N^{\frac{1}{\lfloor dim/2 \rfloor}} \right) + N^{\frac{1}{\lfloor dim/2 \rfloor}}, & \text{if } d < \lfloor \frac{dim}{2} \rfloor \\ N, & \text{otherwise} \end{cases}
\tag{13}
$$

## C  Training algorithm

---

**Algorithm 1** $n$-step PPO with curriculum learning strategy

---

**Input:** initial policy network parameters $\theta$; initial value function parameters $\phi$; clipping threshold $\varepsilon$;
 initial learning rate $\eta_\theta$, $\eta_\phi$, learning rate decay $\beta$.

1: **for** $epoch = 1$ to $E$ **do**
2:    **for** $b = 1$ to $B$ **do**
3:       Randomly generate a batch of training instances $\mathcal{D}_b$;
4:       Initialize random solutions $\{\delta_i\}$ to $\mathcal{D}_b$;
5:       CL: Improve $\{\delta_i\}$ to $\{\delta_i'\}$ by iterating $T_{init} = \frac{S(e) - S(0)}{S(E) - S(0)} \xi^{CL} E$ steps with the current policy
       network (DACT) $\pi_\theta$, where $S(epoch) = 1/\left(1 + e^{-\kappa(epoch - E/2)}\right)$;
6:       Set initial state $s_0 = \{\delta_i'\}$, $t \leftarrow 0$;
7:       **while** $t < T_{train}$ **do**
8:          Collect experience $\{(s_{t'}, a_{t'}, r_{t'})\}_{t'=t}^{t+n}$ by running policy $\pi_\theta$ for $n$ time steps where
          $a_{t'} \sim \pi_\theta(a_{t'}|s_{t'})$;
9:          Set $t \leftarrow t + n$, $\pi_{old} \leftarrow \pi_\theta$, $v_{old} \leftarrow v_\phi$;
10:         **for** $k = 1$ to $K$ **do**
11:            $\hat{R}_{t+1} = v_\phi(s_{t+1})$;
12:            **for** $t' \in \{t, t-1, ..., t-n\}$ **do**
13:               $\hat{R}_{t'} \leftarrow r_{t'} + \gamma \hat{R}_{t'+1}$;
14:               $\hat{A}_{t'} \leftarrow \hat{R}_{t'} - v_\phi(s_{t'})$;
15:            **end for**
16:            Compute PPO-Clip objective $J_{PPO}(\theta)$ using Eq. (14) and clipped value function loss
            $L_{BL}(\phi)$ using Eq. (16);
17:            $\theta \leftarrow \theta + \eta_\theta \nabla J_{PPO}(\theta)$;
18:            $\phi \leftarrow \phi - \eta_\phi \nabla L_{BL}(\phi)$;
19:         **end for**
20:       **end while**
21:    **end for**
22:    $\eta_\theta \leftarrow \beta\eta_\theta$, $\eta_\phi \leftarrow \beta\eta_\phi$;
23: **end for**

---

As presented in Algorithm 1, our training algorithm for DACT is adapted from the proximal policy optimization (PPO) [32], which is a prevailing RL algorithm. In particular, we follow the actor-critic variant of PPO which considers subtracting a baseline $v_\phi(s_t)$ (i.e., value function) in the objective function (line 14) to reduce the variance. Our $v_\phi$ is similar to the one in Wu et al. [11] as follows, (1) it takes the concatenation of node and positional embeddings as input, and then enhances them by a normal multi-head attention layer (with 6 heads); and (2) the enhanced embeddings are passed through a mean-pooling layer (similar to the max-pooling layer in the DAC decoder) and then processed by a four-layer feed forward network (with 128 and 64 hidden units) to get the output value.

We train $\pi_\theta$ and $v_\phi$ for $E$ epochs and $B$ batches per epoch. For each batch, we generate training instances $\mathcal{D}_b$ on the fly (line 3) and use the proposed *curriculum learning* strategy to initialize the state (line 4 to 6). We exploit the $n$-step return estimation to attain a satisfactory trade-off between one step temporal difference (TD) method and Monte Carlo (MC) method [11] (line 8 to 15). Afterwards, PPO performs $K$ epochs of updates on $\mathcal{D}_b$ with its objective clipped by a threshold $\varepsilon$ to penalize large policy variances that move the probability ratio $\frac{\pi_\theta(a_t|s_t)}{\pi_{old}(a_t|s_t)}$ away from 1 (as shown in Eq. (14)),

$$J_{PPO}(\theta) = \frac{1}{|\mathcal{D}_b|n} \sum_{\mathcal{D}_b} \sum_{t'=t}^{t+n} min\left(\frac{\pi_\theta(a_{t'}|s_{t'})}{\pi_{old}(a_{t'}|s_{t'})}\hat{A}_{t'}, \; clip\left[\frac{\pi_\theta(a_{t'}|s_{t'})}{\pi_{old}(a_{t'}|s_{t'})}, 1 - \varepsilon, 1 + \varepsilon\right]\hat{A}_{t'}\right). \quad (14)$$

We also clip the estimated value $\hat{v}(s_t)$ around the previous one as shown in Eq. (15) for better performance [43] and define the baseline loss in Eq. (16). The parameters of our two networks are updated by the Adam Optimizer (line 17,18) with a decaying learning rate (line 22),

$$v_\phi^{clip}(s_{t'}) = clip\left[v_\phi(s_{t'}), v_{old}(s_{t'}) - \varepsilon, v_{old}(s_{t'}) + \varepsilon\right], \quad (15)$$

$$L_{BL}(\phi) = \frac{1}{|\mathcal{D}_b|n} \sum_{\mathcal{D}_b} \sum_{t'=t}^{t+n} max \left( v_\phi(s_{t'}) - \hat{R}_{t'} \;,\; v_\phi^{clip}(s_{t'}) - \hat{R}_{t'} \right)^2. \qquad (16)$$

For our curriculum learning strategy, we adopt $\kappa = 0.2$ and adjust the maximum CL step limit coefficient $\xi^{CL}$ according to the difficulty level and the problem sizes of the routing problems. Ideally, the selected $\xi^{CL}$ should satisfy the following conditions: (1) it is able to considerably boost the sample efficiency of training compared with the smaller ones, and (2) if a larger value is adopted, it may not be able to further bring a significant improvement. In practice, we recommend to determine the value by performing preliminary short training (around 10 epochs) with different $\xi^{CL}$.

## D  Problem-specific description

### D.1  Travelling salesman problem (TSP)

**Problem setup.**  A TSP instance considers to find the shortest loop that visits $N$ nodes exactly once, and finally returns to the original one. We follow Kool et al. [5] to generate the coordinates of $N$ nodes in the unit square $[0, 1] \times [0, 1]$ with a uniform distribution.

**State feature representations.**  The node feature $x_i$ of node $i$ in the state is represented by its location. Let $c_i$ denote the 2-dim coordinates of node $i$. We define $x_i = c_i$.

### D.2  Capacitated vehicle routing problem (CVRP)

**Problem setup.**  On the basis of TSP, we add another node with index $0$ as the depot, and let the original $N$ nodes be customers. A CVRP instance considers to find the minimum total travel distance to serve all customers with multiple vehicles. Hence the solution to CVRP may consist of multiple sub-routes, each of which is a loop of nodes visited by one vehicle, i,e., departing from the depot, visiting the customers on this sub-route, and finally returning to the depot. The constraints of CVRP include: 1) the total demands of a sub-route cannot exceed the vehicle capacity $Q$, and 2) all customers must be visited exactly once. Similarly, we follow Kool et al. [5] to generate the coordinates of all nodes in the unit square $[0, 1] \times [0, 1]$ with the uniform distribution, and sample the demand of each customer uniformly from $\{1, 2, ..., 9\}$. The $Q$ is set to be 30, 40 and 50 for CVRP20, CVRP50 and CVRP100, respectively.

The length of CVRP solution might be larger than $N+1$ since the depot could be visited for multiple times. It may also vary even for the same instance, since different solutions may contain different numbers of sub-routes. For example, both $\delta_0 = \{0, 1, 2, 0, 4, 3, 0\}$ and $\delta_1 = \{0, 1, 2, 3, 4, 0\}$ with different lengths could be feasible solutions to CVRP with 4 customers. Such varying lengths render it much hard for parallel batch training. We thus add multiple dummy depots to the end of initial solutions following Wu et al. [11], where the number of dummy depots could be also considered as the maximum number of available vehicles. In the aforementioned example, after adding dummy depots (we index the depots as (1),(2),(3)), the solution $\delta_1' = \{0^{(1)}, 1, 2, 3, 4, 0^{(2)}, 0^{(3)}\}$ will have the same length with $\delta_0$. In doing so, 1) it guarantees the same length of solutions for a instance batch; and 2) it allows the policy to automatically learn the number of sub-routes and their lengths in a solution. E.g., $\delta_1' = \{0^{(1)}, 1, 2, 3, 4, 0^{(2)}, 0^{(3)}\}$ at step $t$ could be changed to $\{0^{(1)}, 1, 2, 0^{(2)}, 4, 3, 0^{(3)}\}$ (equivalent to $\delta_0$) at step $t+1$ using the 2-opt operator given action $(3, 0^{(2)})$. In our experiments, we empirically set 10 (dummy) depots for CVRP20, and 20 (dummy) depots for CVRP50 and CVRP100, respectively.

**State feature representations.**  For CVRP, the node feature $x_i$ for node $i$ in the state is represented as a 7-dimensional vector [11], which contains, 1) the 2-dim coordinates of node $i$, i.e., $c_i$; 2) the distance from node $i$ to its preceding neighbour; 3) the distance from node $i$ to its succeeding neighbour; 4) sum of demands of the corresponding route before node $i$; 5) the demand of node $i$; and 6) sum of demands of the nodes preceding node $i$ in the same sub-route. Due to the circularity and symmetry of VRP solutions, we consider the succeeding neighbour of the last node in a solution to be the first node, and the preceding neighbour of the first node to be the last node.

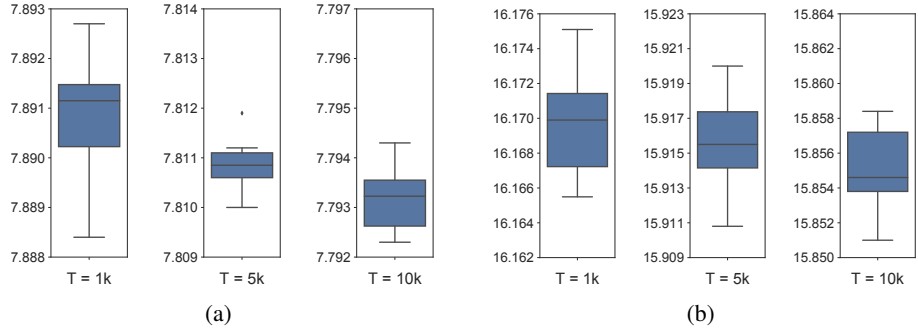

(a)                            (b)

Figure 9: Box plots of the objective values obtained by our DACT model (without augments) for 10 independent runs of 10,000 testing instances (with random seeds 1-10). (a) TSP100; (b) CVRP100.

## E    More discussion on the experiments

### E.1    Hyperparameters

We set $\xi^{CL}$ as 0.25, 2, 10 for TSP 20, TSP50, and TSP100; 1, 4, 12.5 for CVRP20, CVRP50, and CVRP100, respectively. To avoid exploding gradients, we follow [5, 7, 17] to clip the gradient norm of each model parameter to be within 0.04, 0.2, and 0.45 for both TSP and CVRP of the three sizes, respectively. The reward discount factor is set to be $\gamma = 0.999$ for both problems.

### E.2    Training

We train our model with $E = 200$ epochs and $B = 20$ batches per epoch with batch size 600 for TSP and CVRP. We use 512 for CVRP100 due to the limited GPU memory. Regarding the $n$-step PPO algorithm, we set $n = 4$, $T_{train} = 200$ for TSP, and $n = 10$, $T_{train} = 500$ for CVRP. PPO performs $K = 3$ mini-batch updates per batch with its objective function clipped by a threshold $\epsilon = 0.1$. We adopt the Adam optimizer with a learning rate $\eta_\theta = 10^{-4}$ for $\pi_\theta$ and $\eta_\phi = 3 \times 10^{-5}$ for $v_\phi$, both of which are decayed with $\beta = 0.985$ per epoch for convergence. We use pretrained models for TSP50 and CVRP50 to train TSP100 and CVRP100 for faster convergence, while for others the model is initialized randomly. We enable all state transitions of the MDP and the masking for feasibility of a batch to be performed in parallel on GPU for higher efficiency. Training time varies with problem sizes. An epoch takes 6m (minutes), 9m, and 13m for TSP20, TSP50 and TSP100; 21m, 35m, and 53m for CVRP20, CVRP50 and CVRP100, which are shorter than those reported in Wu et al. [11].

### E.3    Stability analysis of our DACT

We study the stability of our DACT model (without augments) during inference stage. Figure 9 depicts the box plots of objective values on TSP100 and CVRP100 for 10 independent runs of 10,000 testing instances, where the minimum, lower quartile, mean, upper quartile, maximum, and possible outlets of results are depicted. In each sub-plot, we show the results for three step limit $T$, i.e., $T = 1$k, 5k, and 10k as per the settings in Table 1. For $T = 1$k, the range of box plots are within only 0.005 for TSP100 and 0.01 for CVRP100. As the step limit $T$ increases, the range of box plots could be further narrowed. These results show that our DACT model has desirable stability for inference.

### E.4    Generalization on benchmark datasets

We now evaluate the generalization performance of DACT by directly applying the trained models in Section 5 to solve instances from two well-known benchmark datasets, i.e., TSPLIB [38] and CVRPLIB [39], respectively. Note that these instances may follow completely different distributions from ours, such as clustered customer locations, corner depot location, etc. We report the results on instances with size between 50 and 200 for TSPLIB; and size between 100 and 200 for CVRPLIB.

As recorded in Table 4 and 5, we first compare our DACT with Wu et al. [11] in the first group of columns to verify the superior performance of DACT to the existing Transformer based improvement model. In the second group of columns, we report the performance of several strong baselines

Table 4: Generalization of DACT v.s. baselines on TSPLIB benchmark dataset.

| Instance | Wu et al. [11] (T=3k) | DACT (T=3k) | OR-Tools | AM-sampling (N=10k) | POMO ×8 augment | Wu et al. [11] (T=3k, M=1k) | DACT (T=10k) | DACT ×4 augment |
|---|---|---|---|---|---|---|---|---|
| eil51 | 2.82% | 1.64% | 2.35% | 2.11% | 0.00% | 1.17% | 0.00% | **0.00%** * |
| berlin52 | 6.34% | 0.03% | 5.34% | 1.67% | 0.03% | 2.57% | 0.03% | **0.03%** * |
| st70 | 4.59% | 0.44% | 1.19% | 2.22% | 0.30% | 0.89% | 0.30% | **0.30%** * |
| eil76 | 6.88% | 2.42% | 4.28% | 3.35% | **1.49%** * | 4.65% | 2.04% | 1.67% |
| pr76 | 1.40% | 1.02% | 2.72% | 2.84% | 19.97% | 1.37% | 0.03% | **0.03%** * |
| rat99 | 17.18% | 4.05% | 1.73% | 9.50% | 7.51% | 8.51% | 1.16% | **0.74%** * |
| KroA100 | 18.39% | 0.86% | 0.78% | 79.49% | 4.45% | 2.08% | 0.63% | **0.45%** * |
| KroB100 | 19.97% | 0.27% | 3.91% | 9.30% | 5.83% | 5.78% | 0.25% | **0.25%** * |
| KroC100 | 22.14% | 1.06% | 4.02% | 8.04% | 6.55% | 3.17% | 0.84% | **0.84%** * |
| KroD100 | 16.33% | 3.54% | 1.61% | 10.02% | 8.74% | 5.00% | 3.54% | **0.12%** * |
| KroE100 | 21.91% | 2.17% | 2.40% | 3.10% | 5.97% | 3.29% | 1.95% | **0.32%** * |
| rd100 | 0.06% | 0.08% | 3.53% | 1.93% | 0.00% | 0.06% | 0.06% | **0.00%** * |
| eil101 | 4.61% | 3.66% | 5.56% | 3.97% | **2.07%** * | 4.61% | 3.66% | 2.86% |
| lin105 | 26.53% | 3.41% | 3.09% | 32.13% | 12.00% | 2.48% | 3.35% | **0.69%** * |
| pr107 | 19.76% | 5.86% | 1.74% * | 43.26% | 5.66% | 3.87% | 5.01% | **3.81%** |
| pr124 | 11.82% | 1.56% | 5.91% | 4.41% | **0.29%** * | 2.97% | 1.22% | 1.22% |
| bier127 | 20.65% | 4.08% | 3.76% | **1.71%** * | 60.56% | 3.48% | 3.79% | 2.46% |
| ch130 | 16.53% | 6.63% | 2.85% | 2.96% | **0.25%** * | 4.89% | 5.48% | 1.93% |
| pr136 | 9.14% | 5.54% | 5.62% | 4.90% | **1.06%** * | 6.33% | 5.14% | 4.54% |
| pr144 | 21.30% | 3.44% | 1.28% | 8.77% | **0.80%** * | 1.40% | 3.44% | 2.49% |
| ch150 | 21.26% | 3.60% | 3.08% | 3.45% | **0.83%** * | 3.55% | 3.45% | 1.23% |
| KroA150 | 17.80% | 6.93% | 4.03% | 9.98% | 13.15% | 4.51% | 3.91% | **3.91%** * |
| KroB150 | 20.20% | 6.10% | 5.52% | 9.87% | 11.72% | 5.40% | 4.10% | **2.82%** * |
| pr152 | 16.20% | 4.48% | 2.92% | 13.47% | 4.11% | **2.17%** * | 3.59% | 3.59% |
| u159 | 21.97% | 6.84% | 8.79% | 7.38% | **2.19%** * | 7.67% | 5.86% | 3.16% |
| rat195 | 25.40% | 6.93% | 2.84% * | 16.57% | 29.06% | 9.90% | 5.81% | **4.99%** |
| d198 | 13.83% | 12.27% | 1.16% * | 331.58% | 45.98% | **4.99%** | 10.74% | 8.75% |
| KroA200 | 22.44% | 3.60% | 1.27% | 15.64% | 20.00% | 7.01% | 1.52% | **1.25%** * |
| KroB200 | 23.69% | 10.51% | 3.67% * | 18.54% | 21.06% | 7.05% | 6.28% | **5.66%** |
| Avg. Gap for [50,100) | 6.53% | 1.60% | 2.93% | 3.61% | 4.88% | 3.19% | 0.59% | **0.46%** * |
| Avg. Gap for [100,150) | 9.69% | 3.01% | 3.29% | 15.29% | 8.16% | 3.53% | 2.74% | **1.57%** * |
| Avg. Gap for [150,200] | 12.76% | 6.81% | 3.70% | 47.39% | 16.45% | 5.81% | 5.03% | **3.93%** * |
| Avg. Gap for all instances | 15.56% | 3.90% | 3.34% | 22.83% | 10.06% | 4.17% | 3.01% | **2.07%** * |

[1] **Bold** indicates that the corresponding method is the best among all learning based ones.
[2] * indicates that the corresponding method is the best among all compared ones.

including, 1) OR-Tools [37], 2) AM-sampling [5], 3) POMO ×8 augment [8], the state-of-the-art neural construction solver, and 4) the enhanced variant of Wu et al. [11], which samples $M$ actions to produce multiple solutions at each step and retrieves the best one as the next state. We present the performance of our DACT with and without augments in the last group of columns. For TSPLIB, we infer the first 5 instances (size < 100) using DACT model trained on TSP50 and the remaining ones using model trained on TSP100. For CVRPLIB, we infer all instances using DACT model trained on CVRP100 since all the sizes are larger than 100. For AM-sampling and POMO, we use the trained models of our sizes which are provided by the authors. The results of Wu et al. [11] and OR-Tools are adapted from Wu et al. [11]. The gaps are calculated based on the optimal solutions provided in the datasets. We also list the average gaps for instances in different problem size intervals, i.e., [50, 100), [100, 150) and [150, 200] for TSPLIB; and [100, 150) and [150, 200] for CVRPLIB.

**TSPLIB**   Pertaining to TSPLIB in Table 4, our DACT (T=3k) significantly outperforms Wu et al. [11] (T=3k) for all instances expect for 'rd100'. It also performs better than the two neural construction solvers AM-sampling (N=10k) and POMO×8 augment for all three problem size intervals in terms of the average gaps, where the superiority to them becomes more obvious as the problem size increases. With larger steps (T=10k), our DACT continues improving the solution qualities and outstrips all the baselines including OR-Tools and Wu et al. [11] (T=3k, M=1k), in terms of the overall average gap. Further empowered by 4 augments, our DACT consistently reduces the gaps and achieves the best performance on most instances with the lowest overall average gap, i.e., 2.07%.

**CVRPLIB**   Pertaining to CVRPLIB in Table 5, the depot and the customers follow various distributions. Though trained on uniform distribution, our DACT (T=5k) outperforms Wu et al. [11] (T=5k), AM-sampling (N=5k), and POMO×8 augment in terms of gap on all instances. Compared with OR-Tools, it yields lower overall average gap and lower average gaps for two problem size intervals. With T=10k and 6 augments, our DACT further reduces the overall average gap to 3.41%. Although POMO×8 augment and Wu et al. [11] (T=5k, M=100) win on 5 C (Clustered) typed instances, our DACT exhibits better performance on other C typed instances. It also outperforms other baselines on

| Instance | Depot Type[1] | Customer Type[2] | Wu et al. [11] (T=5k) | DACT (T=5k) | OR-Tools | AM-sampling (N=10k) | POMO ×8 augment | Wu et al. [11] (T=5k, M=100) | DACT (T=10k) | DACT ×6 augment |
|---|---|---|---|---|---|---|---|---|---|---|
| X-n101-k25 | R | R | 7.70% | 2.09% | 6.57% | 32.95% | 3.64% | 5.60% | 1.86% | **1.47%** * |
| X-n106-k14 | E | C | 4.86% | 2.93% | 3.72% | 6.78% | **1.85%** * | 2.83% | 2.75% | 1.87% |
| X-n110-k13 | C | R | 6.39% | 1.43% | 7.87% | 3.15% | 2.05% | 4.40% | 0.87% | **0.13%** * |
| X-n115-k10 | C | R | 13.32% | 3.29% | 4.50% | 7.52% | 3.49% | 5.19% | 3.26% | **1.68%** * |
| X-n120-k6 | E | RC | 16.16% | 3.50% | 6.83% | 4.54% | **2.12%** * | 5.56% | 3.20% | 2.38% |
| X-n125-k30 | R | C | 8.79% | 6.51% | 5.63% | 35.16% | 7.14% | **4.71%** * | 5.47% | 5.44% |
| X-n129-k18 | E | RC | 11.01% | 2.93% | 8.37% | 4.00% | **0.97%** * | 4.63% | 2.55% | 2.55% |
| X-n134-k13 | R | C | 16.06% | 6.98% | 21.61% | 20.13% | 4.22% | 8.88% | 5.56% | **2.63%** * |
| X-n139-k10 | C | R | 14.99% | 2.54% | 12.02% | 4.30% | 2.28% | 4.90% | 2.16% | **2.08%** * |
| X-n143-k7 | E | R | 20.20% | 7.80% | 11.27% | 8.88% | **2.79%** * | 6.61% | 6.47% | 3.55% |
| X-n148-k46 | R | RC | 16.38% | 2.69% | 7.80% | 79.53% | 19.88% | 3.60% | 2.22% | **2.22%** * |
| X-n153-k22 | C | C | 22.94% | 11.06% | 8.01% | 78.11% | 12.16% | **4.53%** * | 9.02% | 6.53% |
| X-n157-k13 | R | C | 17.15% | 4.64% | 2.57% * | 16.30% | **2.79%** | 3.60% | 4.44% | 3.12% |
| X-n162-k11 | C | RC | 19.16% | 4.43% | 6.31% | 6.37% | 4.77% | 5.26% | 3.04% | **2.62%** * |
| X-n167-k10 | E | R | 18.52% | 5.37% | 9.34% | 8.41% | 4.05% | 8.27% | 4.28% | **3.47%** * |
| X-n172-k51 | C | RC | 12.06% | 6.23% | 10.74% | 85.37% | 21.99% | 4.36% | 5.27% | **3.41%** * |
| X-n176-k26 | E | R | 19.49% | 10.29% | 8.99% | 20.39% | 10.27% | 6.16% | 8.07% | **5.93%** * |
| X-n181-k23 | R | C | 6.27% | 3.41% | 2.94% | 6.45% | 2.08% | 2.08% | 2.42% | **2.08%** * |
| X-n186-k15 | R | R | 17.71% | 5.99% | 7.75% | 6.01% | **2.15%** * | 7.65% | 5.30% | 4.94% |
| X-n190-k8 | E | C | 18.64% | 7.97% | 6.53% * | 46.61% | 9.25% | 6.78% | 6.73% | **6.73%** |
| X-n195-k51 | C | RC | 17.04% | 7.00% | 13.76% | 79.26% | 9.23% | 4.47% | 4.54% | **4.36%** * |
| X-n200-k36 | R | C | 9.60% | 5.93% | 4.15% * | 26.25% | 5.01% | **4.26%** | 5.87% | 5.86% |
| Avg. Gap for [100,150) | | | 12.35% | 3.88% | 8.74% | 18.81% | 4.58% | 5.17% | 3.31% | **2.36%** * |
| Avg. Gap for [150,200] | | | 16.24% | 6.57% | 7.37% | 34.50% | 7.61% | 5.22% | 5.36% | **4.46%** * |
| Avg. Gap for all instances | | | 14.29% | 5.23% | 8.06% | 26.66% | 6.10% | 5.20% | 4.33% | **3.41%** * |

[1] There are three types of depot positions: Central (C), Eccentric/Corner (E), and Random (R).
[2] There are three types of customer distributions: Random (R), Clustered (C), and the mixture of Random and Clustered (RC).
[3] **Bold** indicates that the corresponding method is the best among all learning based ones.
[4] * indicates that the corresponding method is the best among all compared ones.

most of R (Random) typed and RC (mixture of Random and Clustered) typed instances. This further verifies that our DACT generalizes well on real-world instances with various sizes and distributions.

**Remarks** It is worth noting that although POMO×8 augment previously achieved the state-of-the-art performance on synthetic instances according to Kwon et al. [8], it is still lacking in generalization on benchmark instances, whose underlying core model is AM (which showed the worst generalization performance in Table 1). Meanwhile, we can also infer from the results that the neural improvement models including Wu et al. [11] and our DACT have much better generalization capability than the neural construction ones including AM-sampling and POMO. Given the advantages of our DACT model, it achieves the new state-of-the-art generalization performance among all existing Transformer based models on these benchmark instances from TSPLIB and CVRPLIB.

## E.5 Licenses for used assets

We list the used existing assets in Table 6. All of them are open-sourced assets for academic usage.

Table 6: List of used assets and the licenses.

| Asset | Type | License |
|---|---|---|
| OR-Tools [37] | Code | Apache License, Version 2.0 |
| LKH, LKH3 [28, 36] | Code | Available for academic research use |
| AM [5] | Code | MIT License |
| Wu et al. [11] | Code | MIT License |
| POMO [8] | Code | MIT License |
| TSPLIB [38] | Dataset | Available for any non-commercial use |
| CVRPLIB [39] | Dataset | Available for academic research use |
| Python | Code | Python Software Foundation License |
| matplotlib | Code | Python Software Foundation License |
| numpy | Code | BSD License |
| PyTorch | Code | BSD License |
| cv2 | Code | BSD License |
| tdqm | Code | MIT License |
| tensorboard | Code | Apache License 2.0 |