# OpenReview forum: "Learning to Iteratively Solve Routing Problems with Dual-Aspect Collaborative Transformer"
_NeurIPS.cc/2021/Conference — NeurIPS 2021 Poster_

### Official Review · Reviewer_rrox · 2021-07-06

**Rating:** 6
**Confidence:** 4

**Summary:**

The authors introduce three new ideas for the solution of vehicle routing problems. The first is to separate the positional encodings of nodes (in terms of position in a tour) from other node features, such as customer demand or absolute position. For the positional encoding they introduce a handcrafted feature embedding, which they show exhibits cyclic and reflectional symmetry. The second idea is a “dual-aspect collaborative transformer”, which is a pair of attention modules, acting on the positional encodings and node features separately, where the attention weights from the one module can be applied to the value-embedding of the other and vice versa. The third idea is to use curriculum learning in the training process, so that high-quality solutions are also contained in the training set.

The results are competitive with SOTA.

**Ethical Concerns:**

I see no ethical discussion in the paper, I think the authors should include the impact that solving routing problems has on society.

**Limitations And Societal Impact:**

The authors have not included limitations of their work. I think they should address issues of scalability (attention scales quadratically in number of cities), speed (it is still slower than some classical commercial tools), and ease of adapting this framework to new versions of routing problems, such as non-Euclidean/asymmetric TSP.

**Main Review:**

I would like to thank the authors for an interesting read.

My review is marginally below accept because while the methods seem to be new and are competitive, the paper is very hard to read and it is not very clear how or why the methods work from the text. I look forward to the rebuttal, which will hopefully clear up some of the misunderstanding for me.

- Originality

I found the work to be original for the given context. The innovations were as follows:

1) the introduction of a handcrafted feature
2) a new kind of attention mechanism (DACT)
3) the introduction of curriculum learning to the solution of combinatorial optimization problems

- Quality

While the results are appealing, and the methods are new in this context, I think the major drawback of this work is the presentation. The motivation of why the positional and node features should be separated is unclear. It appears that the arguments for doing this may even been a bit obscure to the authors. That’s fine, but I would like to see the text contain some clarity as to what appears to be a pure empirical finding.

Ablation studies are carried out, which confirm the effectiveness of the dual-aspect representation, cyclic positional encodings, and the curriculum learning. This is nice to see and increases the quality of the work.

- Clarity

I found the manuscript readable, but in certain areas quite difficult to understand, owing to the use of terminology, which had hitherto not been defined. It may be advisable for the authors to have the submission checked for readability and grammar.

- Significance

The results appear to be good, in comparison to other methods. That said, it appears that one relevant work in missing. “Deep Policy Dynamic Programming for Vehicle Routing Problems” by Kool et al. (2021) (released February) reports 0.004% optimality gap on TSP100 and -0.127% gap on CVRP with respect to LKH. These numbers are better than the submitted paper, which reports 0.09% and 0.19%, respectively. Nonetheless, the reported numbers are highly competitive.

In terms of ideas, I think the use of curriculum learning is something I have not seen in the context of combinatorial optimization, which makes a lot of sense, thus this is novel. As to the use of the feature embedding and the DACT, these are also novel. The feature embedding is reminiscent of methods from the equivariance literature.


- Questions and suggestions for authors:

Line 36: what is meant by “it can bring mixed correlations and induce noisy biases in the self-attention module”. This sentence is vague for me

Section 2, Positional encoding in Transformer: It may be useful to explain to the reader the relative [17] and united [16] PE methods in more detail, perhaps including mathematical descriptions. This would underscore your contribution better, I feel.

Line 140: I would like a concrete explaination/description/definition of an ‘aspect’.

Line 142: What is ‘correlation noise’?

Section 4.1 PFEs: What is the motivation to choose this particular embedding. Are there other positional embeddings, which would also exhibit cyclic/reflection symmetry?


**Time Spent Reviewing:**

3

---

> ### Author Response · Authors · 2021-08-09
> **Response to Reviewer #rrox**
>
>
> Thanks for the valuable and constructive feedback. We are excited that the reviewer finds our contributions making sense and novel. Regarding the concern for the presentation, we apologize for the confusion caused although other reviewers highly appreciate it. In the response below, we provide concrete definitions and clarifications of some terminology we used, following which we address other raised concerns one by one.
> ***
> **[Definitions/Clarifications of Terminology/Sentence.]**
> 1. **Aspect.** **[Definition]**
> The term “aspect” refers to a particular type of information or feature to describe a VRP solution. **[Clarification]**
> In our paper (as stated in lines 103 to 106), we consider two aspects: 1) problem-specific node features, e.g., coordinations, customer demand, etc; and 2) positional features, i.e., the position/indices of the node in a solution.
> 2. **Mixed Correlations.** **[Definition]** The term “correlation” refers to the dot product between Query and Key in the self-attention module. The term “mixed correlation” refers to the case when the Query and Key are projected from different types of embeddings. **[Clarification]** In our paper, we have two types of embeddings, i.e., node embeddings and positional embeddings. By fusing them into one unified set of embeddings through an addition operator in the original Transformer, there will be four implicit correlations when computing the self-attention in the encoder, i.e.,  node-to-node, position-to-position, node-to-position, and position-to-node correlations (Eq. 9 in Appendix). We refer to the node-to-position and position-to-node correlations as mixed correlations.
> 3. ***“it can bring mixed correlations and induce noisy biases in the self-attention module”*** **[Clarification]** We mean, “the mixed correlations existing in the self-attention calculation can bring unreasonable noises and random biases to the encoder”.
>
> We will add these definitions and clarifications accordingly in the revised paper.
> ***
> **[Motivation for separating the positional and node features.]** We apologize for the confusion. Our major motivation is to question, *“is the original Transformer really good for learning improvement heuristics for VRP?”* With this in mind, we analyzed the differences between NLP tasks (where Transformer originated from) and VRP tasks, based on which we proposed our contributions including 1) DACT, and 2) the CPE. Regarding the former, we separate the positional and node features aiming to address two concerns:
> 1. we should not induce mixed correlations in the design of the encoder (line 34). In Appendix A, we show through empirical study that by fusing the two types of embeddings into a single one, the mixed correlations existing in the self-attention module will bring unreasonable noises and random biases to the encoder;
> 2. we should be cautious on how to encode these two types of heterogeneous and even incompatible information (line 37). For NLP, the order of words is deterministic and instructive, which could help the neural networks better learn the sentence meanings. However, for VRP, the order of customer nodes is not deterministic, and sometimes random (for initial solutions). Thus the fusion of such two information into one set of embeddings may cause incompatible or conflicting correlations in the encoder.
>
> Based on these concerns, the dual aspect representation separates the encoding stream for the two types of information and achieves the collaboration of them through cross-aspect referential attention, the effectiveness of which is verified by the ablation study. Together with other contributions (CPE and CL), we improved the (inference and generalization) performance of neural improvement solvers. We will modify our paper to make this more clear and explicitly state whether each finding is a pure empirical finding or not.
> ***
> **[Why choose PFEs; Motivation and Novelty of CPE.]**
> We would like to clarify that the positional feature embedding (PFEs) is a set of learnable embeddings that we proposed to process the positional features. They are initialized by the CPE (we proposed) and are then enhanced through DACT encoders. Please kindly refer to our general response to all reviewers on the novelty of CPEs. The motivation of CPE is that existing positional encoding methods in NLP are proposed for encoding linear sequences which may not accurately capture the symmetric and cyclic features of VRP solutions.
> ***
> **[Are there any other positional embeddings exhibiting cyclic/reflection symmetry?]**
> According to our general response to all reviewers above, the absolute PE method can indeed capture the cyclic symmetry properties of VRP after training; however, the ability to capture such properties significantly drops when generalizing over different problem sizes. This is a serious drawback since it hinders practical application. The relative PE method seems to be able to extract cyclic features, however, it uses trainable biases and requires fusing the positional and node features together which is found not working for VRPs in [10] (footnote 2 of page 2).  As far as we know, there is no existing work proposing other types of positional encoding for Transformer which is able to accurately encode the cyclic sequences like the routes of VRPs. In contrast, our proposed CPE method enables the Transformer model to encode cyclic sequences, which also helps to significantly improve the generalization performance of neural improvement VRP solvers.
> ***
> **[Limitations of our work.]**
> Regarding *“The authors have not included limitations of their work.”*, please kindly note that we did mention the limitations including 1) our work is more useful for learning improvement models at present in line 345, and 2) the run time of our method still needs to be improved in lines 309 to 314. Regarding the limitations mentioned by the reviewer, we respond as follows:
> * **[Scalability]** The reviewer is correct that the attention indeed scales quadratically in the number of nodes. However, this is also the case for all existing neural VRP solvers based on the self-attention mechanism, including Wu et al. [10], AM [6], POMO[9], CVAE-Opt [12], NLNS [25], L2I [11], etc. This could be alleviated by some techniques such as sparse attention. While the main focus of this paper is not in addressing this, we do agree with the reviewer that scalability is very important, and will certainly work on improving the scalability in the future.
> * **[Speed]** As described in 5.1 (running time analysis), though DACT consumes longer time than LKH on TSP, it consumes shorter time than LKH on CVRP, which suggests that our model is more efficient than this sophisticated solver on more complicated routing problems. Meanwhile, the implementation of LKH is highly optimized and written in C while ours is written in Python. Our model could be further sped up using more efficient implementation and advanced techniques such as TensorRT.
> * **[Ease of adapting this framework to new variants]** Firstly, please kindly refer to our general response to all reviewers on how to adapt our framework to other routing problems. Regarding the non-Euclidean/asymmetric TSP, there is no issue with applying our framework because these TSPs will only affect the way of computing the objective values, and our contributions including CPE, dual-aspect representation, and CL strategy will still work as usual.
> ***
> **[Adding mathematical descriptions for related works.]**
> Thanks for the suggestion, and we will add them.
> ***
> **[Adding reference paper.]**
> Regarding the *“Deep Policy Dynamic Programming for Vehicle Routing Problems”*, we thank the reviewer for offering this impressive concurrent work, and we will definitely include it in our related works.
> ***
> **[The “correlation noise” (line 142).]**
> We apologize for this vague phrase. We will change it into “mixed correlations” so the sentence would become “ … the self-attention scores are computed individually for each aspect to avoid mixed correlations, and …”. And we will thoroughly check and revise other vague sentences.

---

> > ### Comment · Reviewer_rrox · 2021-08-24
> > **Update review: 5 -> 6**
> >
> > Dear authors,
> >
> > Having read through the rebuttal and other reviews I have decided to up my score from 5 to 6.
> >
> > *Readability*: I have upped my score primarily in accordance with my initial major concern about readability. The authors addressed my concerns and have detailed quite thoroughly how they will improve specific areas of ambiguity.
> >
> > *Results*: I think the results are competitive; although, not SOTA. These problems are extremely tough to work on and I think the main contribution of this project is conceptual. As a result, I fill this is satisfactory for publication.
> >
> > *Door to further improvements*: This specific feature representation will probably not be used by follow-up works, but the principle of respecting cyclic shift in/equivariance and other symmetries could be very important. I have not considered this before and this is a further reason to for the 5 -> 6 increase.

---

> > > ### Author Response · Authors · 2021-08-24
> > > **Response**
> > >
> > > We deeply appreciate the support of the reviewer, especially for acknowledging the value of the model in exploiting the cyclic or symmetric property. Meanwhile, we have also tested POMO($\times$8 augment) on the instances of TSPLIB and CVRPLIB (Table 4 and 5 in Appendix), which achieved average gaps of 10.6% and 6.1%, respectively. Those results are inferior to the ones achieved by our DACT in Table 4 and 5, i.e., 2.1% and 3.4%, respectively, which further justified the desirable generalization performance of our model. We will include the new results in the final version, and also further polish our work.

---

### Official Review · Reviewer_wmXs · 2021-07-07

**Rating:** 6
**Confidence:** 4

**Summary:**

The paper proposes a new algorithm for solving the traveling salesman problem and capacitated vehicle routing problem by learning embeddings for the node and positional features separately. The algorithm uses a cyclic positional encoding method (CPE) to characterize the geometric properties of VRP solutions and designs a curriculum learning strategy to improve the sample efficiency. The experiment results show that for TSP and CVRP the proposed algorithm outperforms existing Transformer-based improvement models.

**Limitations And Societal Impact:**

Here are some other comments:
1) The paper shows that CPE captures the cyclic and similarity properties of TSP well (Figure 4). The experiment results demonstrate that the model can also solve the CVRP in relatively good quality. However, we know CVRP has cyclic and symmetric properties within each sub-route. It would be helpful if you can provide some intrinsic illustrations as you did in Figure 4 to show that the structure proposed is also able to deal with the circularity and symmetry of the sub-route for CVRP. If not, do you have any additional ideas to improve the model?
What concerns me is whether the model can be generalized well to more complicated problems such as CVRPTW, which have more geometric properties.
2) When applying CPE, the dimension of the input becomes rather large. It can adversely impact the computational efficiency of the model. Does it potentially constrain the scalability of the model, especially when we need to deal with large-scale problems?
3) For CVRP, the pairwise operator does not always result in a feasible solution. How does the model handle such a situation?

**Main Review:**

Originality: There are two contributions in this work:
1) Learning node and position embedding separately to decrease potential noises,
2) Using the cyclic positional encoding method to handle the circularity and symmetry of VRP solutions,

which I think are novel and interesting.

Quality: The paper is solid. Authors provide both synthetic and real data experiments with detailed analysis.

Clarity: The paper is clearly written and easy to follow.

Significance: This paper proposes a new embedding method to efficiently handle the circularity and symmetry of VRP solutions. The experiments show that the proposed algorithm achieves a better objective value than existing Transformer-based improvement models. However, this algorithm still cannot beat LKH and another baseline [1]. Meanwhile, the computational time seems a bit longer compared with [2] as they have almost the same performance.

[1] Lu, Hao, Xingwen Zhang, and Shuang Yang. "A learning-based iterative method for solving vehicle routing problems." International Conference on Learning Representations. 2020.

[2] Kwon, Yeong-Dae, et al. "POMO: Policy Optimization with Multiple Optima for Reinforcement Learning." Advances in Neural Information Processing Systems 33, 2020.

**Time Spent Reviewing:**

10

---

> ### Author Response · Authors · 2021-08-09
> **Response to Reviewer #wmXs**
>
> Thanks for taking a long time to review our paper and offer the valuable comments.  We are excited that the reviewer finds our work novel and interesting. We understand that the reviewer still has concerns about the final performance of our DACT, the significance and the scalability of our CPE method. In the following we respond to them.
> ***
> **[Performance does not exceed LKH and L2I.]**
> * We do acknowledge that the performance of our DACT did not exceed LKH, which is also the case for latest learning based methods such as POMO [9] and CVAE-opt [12]. We aim to further improve the performance and outperform strong conventional solvers like LKH eventually. For L2I, although it can surpass LKH on CVRP, it is largely due to the fact that they use a big group of hand-crafted local search operators specialized to CVRP, which may limit its application in solving other routing problems. Moreover, L2I takes an extremely long inference time (about 4,000 hours for 10,000 instances, as estimated according to the 24min/instance reported in its original paper).
> * On the other hand, as mentioned in our general response to all reviewers, our method significantly improves the generalization performance over different problem sizes of existing neural improvement heuristics (generalization gaps of around 16% of [10] v.s. our 4% in Table 3, and generalization gaps of 79% and 31% of AM v.s. our 2% and 3% on TSPLIB and CVRPLIB). Compared with the neural construction heuristic AM  (which is also the underlying model of POMO), DACT also delivers superior generalization performance. In particular, the average gaps of AM on TSPLIB and CVRPLIB are 79% and 31% while ours are 2% and 3% (Appendix F.3). Without such good generalization, the existing neural solver may not be practical for use since the VRP instances in reality always have different sizes.
> ***
> **[Adapt CPE for sub-routes and CVRPTW.]**
> We currently add multiple dummy depots and concatenate all sub-routes to form a larger cyclic sequence. For example, if we have sub-route 0,1,2,0 and 0,3,4,5,0, we can form a cyclic sequence of 0,1,2,0,3,4,5 where depot (node 0) is repeated twice in the encoding (we call them two dummy depots in line 562 of appendix). In this sense, node 0 is believed to be the neighbours of both node 1,2 (for sub-route 1) and the neighbours of node 3,5 (for sub-route 2) by CPE. This method is simple and straightforward, which already delivers good performance in our experiments. However, adopting multiple groups of independent CPE to encode each sub-route proposed by the reviewer is very interesting and we will consider this in future work which could further improve the solution qualities. Regarding using our DACT for CVRPTW, please kindly refer to the general response to all reviewers above.
> ***
> **[The scalability of our CPE.]**
> Regarding *“When applying CPE, the dimension of the input becomes rather large. It can adversely impact the computational efficiency of the model”*, it is a misunderstanding since our CPE has exactly the same dimensionality as the original PE of the Transformer. The dimension of input does not change before and after applying CPE. After applying CPE or PE, each node in the solution would be embedded to a 64-dim vector which is the same as the Transformer for NLP where each word in the sentence will be embedded to a 64-dim vector. The major difference is that CPE can make the first node and last node be adjacent, while PE cannot. Meanwhile, the initialization of CPE only needs to be performed once in advance and we can reuse the vectors for different solutions during the subsequent inference. Thus there should be no issue regarding the scalability of CPE. For large-scale instances, please kindly refer to appendix F.3, where we tested DACT on benchmark datasets containing CVRP200 and TSP200 instances.
> ***
> **[Feasibility of solutions.]**
> The reviewer is right. We thus apply feasibility masks in the decoder (line 228 in the paper) to mask the infeasible actions, similar to AM, POMO and Wu et. al. [10]. In our implementation, feasibility masks are calculated in parallel for a batch.

---

> > ### Comment · Reviewer_wmXs · 2021-08-24
> > **Response**
> >
> > Thank you very much for your reply, especially for the detailed explanation of the scalability of CPE.

---

> > > ### Author Response · Authors · 2021-08-25
> > > **Response to response**
> > >
> > > Thanks for the acknowledgment and support.

---

### Official Review · Reviewer_9zdA · 2021-07-16

**Rating:** 5
**Confidence:** 4

**Summary:**

Simply, this paper adds the following to an existing paper titled "Learning improvement heuristics for solving routing problems" [Ref 10].

1. Dual aspect representation: a typical neural architecture is converted to a dual. Problem encoding and its solution (the order of routing) encoding are separately done
2. Cyclic positional encoding
3. Curriculum learning

**Limitations And Societal Impact:**

Yes

**Main Review:**


In Figure2: DAC Decoder output, P_1N is misrepresented by P_11.

One of the strong points of the paper is that node embedding and positional embedding can be mixed (the Transformer arch.). For a routing problem, the standard benchmark results prove a good performance. Taking a simplistic neural architecture, the authors achieve new state-of-the-art  for the benchmark TSP and CVRP problems. Particularly, for CVRP-20, the proposed approach beats LKH.

Cyclic positional encoding is useful for cyclic routing problems, although, for more general routing problems can't be helped as much.

241: in curriculum learning strategy, the authors should elaborate more in "gradually prescribes higher-quality solutions as the initial states for training." (Appendix D has detailed explanations, but the main content should explain the key idea and why.

287: why need to record time for the case of multiple GPU card? Isn't recording single GPU time fair or not?

In Table 1, an existing approach by Wu et al. [10] and new model performances are compared. Why for using the same 1 GPU the inference time differences are significant?

If the authors conduct an ablation study and include, the paper will get better.


**Time Spent Reviewing:**

2

---

> ### Author Response · Authors · 2021-08-09
> **Response to Reviewer #9zdA**
>
> Thanks for the valuable feedback. We are excited that the reviewer acknowledges the performance of our method. We understand that the concern is mainly regarding the effectiveness and the significance of our CPE method, for which we respond as follows.
> ***
> **[Significance of the CPE.]**
> First, please kindly refer to our general response to all reviewers on this point and the extension to other VRPs. Besides, we would like to seek the reviewer’s understanding that VRP is a family of NP-hard problems with hundreds of variants, and it could be too ambitious to propose a generic technique that works well for all of them. Instead, we focus on handling the cyclic property which widely exists in many practical routing problems such as SDVRP, PCTSP, PDP, CVRPTW. CPE is not a silver bullet for all VRPs, but the benefit of it has been demonstrated on two representative benchmark problems, i.e., TSP and CVRP.
>
> ***
>
> **[Explanations of curriculum learning (CL) strategy.]**
> We would like to clarify that Appendix D only contains the detailed equations and explanation on how we perform CL strategy, and we use lines 234 to 245 (main paper) to explain the key idea and why. The reason for CL strategy is that high-quality solutions are usually not observed during training by the agent in existing methods (line 237) so that the estimations for future reward by value functions (critic network) may be of high variance (line 238). Therefore, we aim to seek a simple yet effective solution to tackle this. The key idea in our design is that we could use higher-quality solutions as the initial states to force the network to observe high-quality solutions (line 242). Given the suggestion from curriculum learning [32], we need to fulfill this in a gradual manner to avoid instability and achieve better sample efficiency (line 243).  We will accordingly refine the statement in the revised paper.
> ***
> **[Number of GPUs.]**
> We use multiple GPU cards to solve 10,000 testing instances mainly due to the memory limitation of one GPU card, which prevents us from using large batch sizes commonly used in other methods. We believe this is not a critical issue in practice use as currently some GPU cards already have sufficiently large memory, e.g., RTX 3090 has 24GB memory which is fairly enough for our model to solve 10,000 instances using only one card.
> ***
> **[Computation time compared with [10].]**
> The shorter computation time of our DACT originates from the following three aspects:
> 1. Although we capture the dual-aspect information (NFEs and PFEs), we reduce the dimension of the network, where we use 64-dimensional embeddings for both NFEs and PFEs, while [10] uses 128-dimensional embeddings (2 $\times$ 64 $\times$ 64 < 128 $\times$ 128). In doing so, we even have fewer parameters in total than [10] (DACT has around 0.29M parameters while the network of [10] in ablation study has around 0.37M parameters);
> 2.  We have enabled efficient implementation and batch parallelism for all state transition and feasibility masks in our codes (will be released on GitHub);
> 3. The hardware for training our DACT and [10] might be different.
>
> In general, the SA-T (which is the model of [10] but with more efficient implementation by us) will be slightly faster than our DACT (e.g., for solving TSP 100, DACT consumes 2.5h while SA-T is about 20 mins faster). Given the significant improvement on the (inference and generalization) performance of our DACT over [10] (table 1), we believe such slight degradation on inference time is acceptable. We will add the parameter/time comparison in the ablation study (Table 2) and explicitly mention the above clarifications in the revised paper.

---

> ### Author Response · Authors · 2021-09-01
> **Discussion**
>
> Dear Reviewer #9zdA,
>
> Thanks for taking the time to review our paper. May we know if you still have additional concerns?

---

### Official Review · Reviewer_YbAL · 2021-07-18

**Rating:** 7
**Confidence:** 5

**Summary:**

This paper introduces a novel architecture for solving the VRP. After illustrating the issues concerning the positional encoding for VRP, this paper introduces an alternative feature representation called CPE which captures the circular dependencies of the VRP. By using this new feature extraction and a specially designed Transformer network to encode the information of nodes and sequences, they show that their method is the best among the baselines and can "solve" VRP to close-to-optimality in a short amount of time. I like the main idea of the paper and how all components are designed to work in synergy. It is very clearly written and easy to follow.

**Main Review:**

This paper includes comprehensive illustrations, ablations studies, and detailed experiments setup discussions, so it does not allow many questions. I do have a few questions/comments to further make this work better:

1) It is not clear what the N nodes demonstrate in this paper in Section 3. While reading the main body, I was thinking that the depot is one of those N nodes, but from line 558 of the Appendix, it appears that the depot is the node with ID N+1.

2) At the algorithm improves the solutions, there might be fewer routes (or vehicles) necessary to satisfy the demands. It is not clear how the algorithm is able to reduce the number of routes. Can you elaborate on this?

3) Another question about the depot. How the location of the depot is going to affect the solution? It looks like that there is no information on the depot is used during the training. This needs to be clarified as it is not obvious how decisions to visit the depot are determined.

4) What will happen in figure 3 if the number of nodes is not a power of two? How we can make sure that the first and last nodes in the gray code are adjacent (different in one digit)? I suggest using the same TSP20 example that you used in Figure 4 to illustrate how this coding works.

5) My next question is about the reward computation. When you defined the reward in Section 3, I was thinking that you have a fixed set of problems (it can be millions of them) and then you have a record of what is the best-observed reward for every specific problem during the training. I thought that is how you get $D(\delta^*), but based on the explanation in Section 5, you generate sample problems "on the fly".  I am wondering how you compute this quantity.

6) There is an argument in line 270 which says "which are shorter than those reported in [10]". I have some doubts about this. Since your model is much larger than [10], how can this happen? Do you have any justification? Or, that is simply due to efficient implementation and parallelism?

7) Why L2I is not included in your baselines? Even if it takes some time, it is the SOTA of learning-based methods so far (as far as I know).

8) You have argued that other operators are not as good as 2-opt. It would be informative to have an ablation study on how changing the improvement operator affects the results.


Other comments:
* It needs to be clarified what are the "preset pairwise operators" in Line 113.
* In the description of Figure 1, instead of using vague terms like "original one" and "ours", you could cite the work that considers (a) and also use the name of your framework (DACT).
* Do you mean state transition in Line 129 (instead of transient)?
* In Figure 2, I think matmul (not matmal) is the is a most common way to denote a matrix multiplication. Also, about the operations after softmax (before concat and linear project), are they matrix multiplication or just a scalar times vector? If the latter is correct, you may use "mul" instead of "matmal".
* This is very minor, but I am wondering why you report the cumulative time for 10000 problems. It makes more sense to me to know what is the expected time to solve a random problem using a method.
* Why did you introduce new notation $pe_i$ instead of g_i inf Appendix A?






**Time Spent Reviewing:**

6

---

> ### Author Response · Authors · 2021-08-09
> **Response to Reviewer #YbAL**
>
> Thanks for taking a long time to review our paper and offer valuable feedback. We are gratified that the reviewer likes the idea of our work. Following the suggestions, we conducted additional experiments and will update the paper accordingly. We hope that the following response will clear the questions.
> ***
> 1. N refers to the number of customers with indices 1,2,...,N. For CVRP, we add a depot with index 0, so N+1 in line 558 means the total number of nodes.
>
> 2. Yes, our algorithm can reduce or increase the number of sub-routes automatically during optimization as explained in lines from 559 to 571 (Appendix). Here we briefly recap the example in line 560 (Appendix). There are two sub-routes in solution $\delta_0 = [0, 1, 2, 0, 4, 3, 0]$ (0, 1, 2, 0 and 0, 4, 3, 0 where 0 is the depot) and one sub-route in solution $\delta_1 = [0, 1, 2, 3, 4, 0, 0]$. The solution $\delta_1$ can be obtained from solution $\delta_0$ by applying 2-opt operation (i.e., reverse the segment ‘0, 4, 3’ to ‘3, 4, 0’).
>
> 3. Regarding the comment “It looks like that there is no information on the depot is used during the training”, it is not true as we treat the depot as a special customer with 0 demand. For example, to encode solution $\delta_0 = [0, 1, 2, 0, 4, 3, 0]$, the location of all these 6 nodes will be encoded where the depot is repeated three times and we call them dummy depots (line 562 in Appendix).
>
> 4. The reviewer is right that the cyclic gray code works only when the number of nodes is a power of two as they are binary codes. However, here the gray codes are only used to illustrate our motivations. Our actual encoding scheme, the cyclic positional encoding (CPE), can support arbitrary lengths larger than 2 since it generates real-valued vectors as shown in Eq (1). In fact, the problem sizes selected in experiments, i.e., 20, 50, and 100, are not powers of two. Kindly refer to Figure 4(a) for visualization of our CPE with respect to 20 nodes.
>
> 5. We apologize for the misunderstanding. In the training process, the problem instances are randomly generated on the fly. During training, each generated batch will be solved by $T_{train}$ iterations. The reward was set to be the immediate reduced cost of the best incumbent solution after each step which ensures the cumulative reward equal to the total reduced cost over the initial solution. We use $\delta^*_t$ referring to the best incumbent solutions found until iteration $t$. Specifically, $\delta^*_0$ is initialized using the randomly generated solutions. Then, in each iteration, the DACT generates a new solution and the $\delta^*_t$ will be updated if this solution is better. We will add a footnote to this.
>
> 6. The shorter computation time of our DACT originates from the following three aspects: 1) Although we capture the dual-aspect information (NFEs and PFEs), we reduce the dimension of the network, where we use 64-dimensional embeddings for both NFEs and PFEs, while [10] uses 128-dimensional embeddings (2 $\times$ 64 $\times$ 64 < 128 $\times$ 128). In doing so, we have even fewer parameters in total than [10] (DACT has around 0.29M parameters while the network of [10] in ablation study has around 0.37M parameters); 2) We have enabled efficient implementation and batch parallelism for all state transition and feasibility masks (codes will be released on GitHub); 3) The hardware for training our DACT and [10] might be different. We will add the parameter/time comparison in the ablation study (Table 2) and explicitly mention the above clarifications in the revised paper.
>
> 7. Note that we have reviewed L2I in the related work section. The reason for not including it as a baseline is because it learns to select from a pool of traditional hand-crafted operators, while our method focuses on automatically learning improvement actions end-to-end for a given operator. Nevertheless, we will follow the suggestion and add L2I to the baselines.
> 8. We will add the ablation study on different improvement operators. From the experimental results, we find that 2-opt > insert > swap.
> ***
> **[Other comments.]** Regarding the “preset pairwise operator”, we mean an operator from 2-opt, swap, and insert which is specified before training. Regarding the computation efficiency, we follow the common settings of existing neural VRP solvers including AM [6], CVAE-OPT [12], POMO [9], and report the computation time of all 10,000 instances, which are solved in several batches in parallel. So the running time will only depend on the number of improvement steps (T) for a given batch of instances and the hardware. For the time of solving one instance, as an example, DACT (T=5k) takes around 3 mins, DACT (T=10k, $\times$ 6 augment) takes around 6 mins to solve a CVRP200 instance from CVRPLIB. For the remaining comments, we have revised our paper accordingly.

---

### Author Response · Authors · 2021-08-09
**General Response To All Reviewers**

First of all, we are greatly thankful to all the reviewers for their valuable comments. We are excited that most of the reviewers found our work novel and interesting. Here, we would like to clarify the significance regarding the cyclic positional encoding (CPE). Hopefully, the response below could clarify the concerns by reviewer #9zdA and #wmXs that CPE may not work for more complicated routing problems (like CVRPTW).
***
**[The significance of our CPE.]**
- One major concern on applying neural VRP solvers into practice is the generalization performance across different problem sizes. Ideally, a model trained on CVRP100 should be able to solve instances of other sizes close to 100 (say from 50 to 150). However, such performance of the original PE is relatively poor, e.g., a gap of 16.37% in the ablation study (Table 3) and 15.56% on TSPLIB (Table 4) while the corresponding gaps of our CPE are 3.87% and 3.9%, respectively.
- In the visualization (Figure 5) and lines 331 to 336, the rationale behind this was also analyzed. We found that even though the original PE method does not consider the cyclic property, the learned embeddings after training still capture this pattern (as highlighted in Figure 5(a)). This is fine when the model is trained and tested on the same size. However, when tested on different sizes, the encoder might get confused and generate random attention scores (see Figure 5(a)).
- In contrast, our CPE method could capture the cyclic property more accurately and hence exhibits better generalization across different sizes. When compared with the construction method, e.g., AM (which is also the underlying model of POMO), DACT also delivers superior performance. In particular, the average gaps of AM on TSPLIB and CVRPLIB are 79% and 31% while ours are 2% and 3% (Appendix F.3).
- Finally, we would like to note that CPE enables applying Transformer to other tasks besides VRP where the cyclic property is important, such as encoding circular DNA/RNA structure in computational biology [39][40].

References:

[39] Liu, Chengyu, et al. "Biogenesis mechanisms of circular RNA can be categorized through feature extraction of a machine learning model." Bioinformatics 35.23 (2019): 4867-4870.

[40] Yu, Chun-Ying, et al. "The circular RNA circBIRC6 participates in the molecular circuitry controlling human pluripotency." Nature communications 8.1 (2017): 1-15.

***
**[Adapting CPE to other VRP problems.]**
First, we would like to clarify that we did not claim that our CPE method works for all types of routing problems. Following POMO [9], CVAE-Opt [12], AM [6], Wu et. al. [10], etc, we focus on the most classic ones, including TSP and CVRP. In the design of CPE, we assume that the solution to the targeted routing problem is a sequence of nodes that starts and ends at the same node, which holds for many other types of routing problems, including SDVRP, PCTSP, PDP, CVRPTW, etc. In other words, we consider the whole solution as a large circle, which is already very helpful especially in generalization as shown in the CVRPLIB experiments (Table 5). We believe that explicitly handling circular property in sub-routes could further boost our method, which is an important future direction. To handle more complicated problems, as long as they satisfy the assumption that the route (solution) is a cyclic sequence, we could apply CPE to learn the Positional Feature Embeddings (PFEs) as usual and learn other problem-specific features using Node Feature Embeddings (NFEs). For example, to adapt our DACT to CVRPTW mentioned by reviewer #wmXs, we could add the starting and ending time of serving each customer to each node feature $x_i$, which is similar to how we add the demand of each customer to $x_i$ for CVRP as described in line 572 to 578 (Appendix). We will add more discussions on this in the revised paper.

---

### Decision · Program_Chairs · 2021-09-28

**Decision:**

Accept (Poster)

**Comment:**

The paper focuses on the architecture of neural end-to-end combinatorial optimization methods for routing problems like the TSP and the CVRP. The authors propose two improvement to the encoder of such problems: (1) encoding the node features and position features separately, which results in better behavior of attention scores and (2) using a cyclic positional encoding, that better capture symmetries of VRP solutions.

The reviews on the paper are mixed. Only one reviewer recommended reject, this reviewer did not react to the author's reply or the other reviewer's arguments. Keeping this in mind, I'd summarize the opinion of the reviewers as follows:
- Originality: Most of the reviewers found the methods to be novel and interesting, although the core method stays close to the established framework of e.g. [10].
- Technical quality: One reviewer mentions the quality is solid. The reviewers seem mostly satisfied about quality, too. One reviewer explicitly mentions the detailed ablations.
- Significance and Relevance: The topic is relevant to the NeurIPS community. The reviewers characterize the results slightly differently, from 'best among baselines' and 'new state of the art' to 'competitive with baselines'/'doesn't beat baselines convincingly'. However, except from the overall quality, the authors study the generalization of the method to larger problems, which does seem convincingly better than that of competitor methods. Since scaling is a major obstacle for neural combinatorial optimization methods, this is a significant benefit.

Overall, I consider the paper to close to the borderline, however, I think the generalization results are quite interesting for the NeurIPS audience and as such I would recommend acceptance.


**Consistency Experiment:**

NeurIPS has a long history of experimentation. In 2014, NeurIPS ran an experiment in which 10% of submissions were reviewed by two independent committees to quantify the randomness in the review process. This year, we repeated a variant of this experiment to see how the quality of the review process has changed over time.  This paper was part of the experiment and was therefore assigned to two committees (consisting of reviewers, an Area Chair, and a Senior Area Chair) that reached independent decisions.  If both committees made the same recommendation, this recommendation was followed. If a single committee recommended acceptance, the paper was accepted (with the exception of a few cases in which the other committee identified what we considered a fatal flaw, e.g., an error in a key result).

Both committees reached the same decision: **Accept (Poster)**

The other committee assigned to the paper recommended **Accept (Poster)**.  You can find the other set of reviews, along with any follow up discussion with the authors here:
https://openreview.net/forum?id=TmLqkYn71gV